# The giant staphylococcal protein Embp facilitates colonization of surfaces through Velcro-like attachment to fibrillated fibronectin

**Nasar Khan[1†], Hüsnü Aslan[1*†], Henning Büttner[2], Holger Rohde[2], Thaddeus Wayne Golbek[3], Steven Joop Roeters[3,4], Sander Woutersen[4], Tobias Weidner[3], Rikke Louise Meyer[1,5]***

[1]Interdisciplinary Nanoscience Center (iNANO), Aarhus University, Aarhus, Denmark; [2]Institute for Medical Microbiology, Virology and Hygiene, University Medical Centre Hamburg-Eppendorf, Hamburg, Germany; [3]Department of Chemistry, Aarhus University, Aarhus C, Denmark; [4]Van 't Hoff Institute of Molecular Sciences, University of Amsterdam, Amsterdam, Netherlands; [5]Department of Biology, Aarhus University, Aarhus C, Denmark

**\*For correspondence:**
asl@dfm.dk (HA);
rikke.meyer@inano.au.dk (RLM)

[†]These authors contributed equally to this work

**Competing interest:** The authors declare that no competing interests exist.

**Abstract** *Staphylococcus epidermidis* causes some of the most hard-to-treat clinical infections by forming biofilms: Multicellular communities of bacteria encased in a protective matrix, supporting immune evasion and tolerance against antibiotics. Biofilms occur most commonly on medical implants, and a key event in implant colonization is the robust adherence to the surface, facilitated by interactions between bacterial surface proteins and host matrix components. *S. epidermidis* is equipped with a giant adhesive protein, extracellular matrix-binding protein (Embp), which facilitates bacterial interactions with surface-deposited, but not soluble fibronectin. The structural basis behind this selective binding process has remained obscure. Using a suite of single-cell and single-molecule analysis techniques, we show that *S. epidermidis* is capable of such distinction because Embp binds specifically to fibrillated fibronectin on surfaces, while ignoring globular fibronectin in solution. *S. epidermidis* adherence is critically dependent on multivalent interactions involving 50 fibronectin-binding repeats of Embp. This unusual, Velcro-like interaction proved critical for colonization of surfaces under high flow, making this newly identified attachment mechanism particularly relevant for colonization of intravascular devices, such as prosthetic heart valves or vascular grafts. Other biofilm-forming pathogens, such as *Staphylococcus aureus,* express homologs of Embp and likely deploy the same mechanism for surface colonization. Our results may open for a novel direction in efforts to combat devastating, biofilm-associated infections, as the development of implant materials that steer the conformation of adsorbed proteins is a much more manageable task than avoiding protein adsorption altogether.

## Editor's evaluation

Bacteria must adhere to tissue to colonize and, in some cases, cause disease. Thus adherence is a key to understanding the pathogenesis of infectious diseases. This study uses a range of microscopic and biophysical measures to discover that Embp, a very long protein of repeating subunits, facilitates adherence of Staphylococcus epidermidis to damaged tissue or foreign bodies (e.g. catheters or implantable devices) even under high flow conditions such as those found in blood.

**eLife digest** A usually harmless bacterium called *Staphylococcus epidermidis* lives on human skin. Sometimes it makes its way into the bloodstream through a cut or surgical procedure, but it rarely causes blood infections. It can, however, cause severe infections when it attaches to the surface of a medical implant like a pacemaker or an artificial replacement joint. It does this by forming a colony of bacteria on the implant's surface called a biofilm, which protects the bacteria from destruction by the immune system or antibiotics.

Understanding how *Staphylococcus epidermidis* implant infections start is critical to preventing them. This information may help scientists develop infection-resistant implants or new treatments for implant infections. Scientists suspect that *Staphylococcus epidermidis* attaches to implants by binding to a human protein called fibronectin, which coats medical implants in the human body. Another protein on the surface of the bacteria, called Embp, facilitates the connection. But why the bacteria attach to fibronectin on implants, and not fibronectin molecules in the bloodstream, is unclear.

Now, Khan, Aslan et al. show that Embp forms a Velcro-like bond with fibronectin on the surface of implants. In the experiments, Khan and Aslan et al. used powerful microscopes to create 3-dimensional images of the interactions between Embp and fibronectin. The experiments showed that Embp's attachment site is hidden on the globe-shaped form of fibronectin circulating in the blood. But when fibronectin covers an implant surface, it forms a fibrous network, and Embp can attach to it with up to 50 Velcro-like individual connections. These many weak connections form a strong bond that withstands the force of blood pumping past.

The experiments show that the fibrous coating of fibronectin on implants makes them a hotspot for *Staphylococcus epidermidis* infections. Finding ways to block Embp from attaching to fibronectin on implants, or altering the form fibronectin takes on implants, may help prevent these infections. Many bacteria that form biofilms have an Embp-like protein. As a result, these discoveries may also help scientists develop prevention or treatment strategies for other bacterial biofilm infections.

## Introduction

Biomedical implants such as catheters, prosthetics, vascular grafts, and similar devices have revolutionized the medical field. However, implants can lead to severe infections due to bacterial biofilms: The formation of multicellular bacterial communities encased in a protective extracellular matrix (*Arciola et al., 2018*). Bacteria in the biofilm evade phagocytosis by immune cells (*de Vor et al., 2020*), and the immune system can therefore not eradicate the infection. Furthermore, a fraction of the cells enter a dormant state in which they are highly tolerant to antibiotics (*Rowe et al., 2021*). With the rise in use of biomedical implants, there is an urgent and growing need to understand how biofilm infections arise, such that new strategies for preventative treatment can be developed.

Staphylococci, particularly *Staphylococcus aureus* and *Staphylococcus epidermidis* are the culprits of most implant-associated infections (*Oliveira et al., 2018*). Despite its low virulence, *S. epidermidis* is common in these infections due to its prowess in biofilm formation. *S. epidermidis* attaches to implant surfaces via adsorbed host proteins (*Patel et al., 2007*; *François et al., 1998*) and it expresses an array of surface-bound proteins (adhesins) that bind to host proteins, such as fibrin, fibronectin (Fn), vitronectin, and collagen to initiate biofilm formation (*Foster, 2020*). One such adhesin is the extracellular matrix-binding protein (Embp), which is found in the vast majority of clinical isolates of *S. epidermidis* (*Rohde et al., 2007*; *Salgueiro et al., 2017*), suggesting that this giant 1 MDa protein is important for this species' pathogenicity. Embp contains a number of repetitive motifs. These were originally described based on sequence similarity to be 21 'Found in Various Architecture' (FIVAR) repeats and 38 alternating repeats of 'G-related Albumin Binding' (GA) and FIVAR repeats combined, named FIVAR-GA repeats (*Figure 1*). After the crystal structure was recently solved, the domain structure was updated and consists of ten 170-aa F-repeats that each represent two FIVAR repeats, and forty 125-aa FG-repeats that each represent the previously termed FIVAR-GA repeats (*Büttner et al., 2020*). These 50 repeats can bind to Fn by interacting with FN12 of the Fn type III repeats (*Figure 1*), and it is presumed that this interaction aids the colonization of the host (*Christner et al., 2010*).

The Fn deposition can occur around implants (*Wolfram et al., 2004*) and offer a site for bacterial attachment. We wondered how bacteria like *S. epidermidis* can colonize implant surfaces by

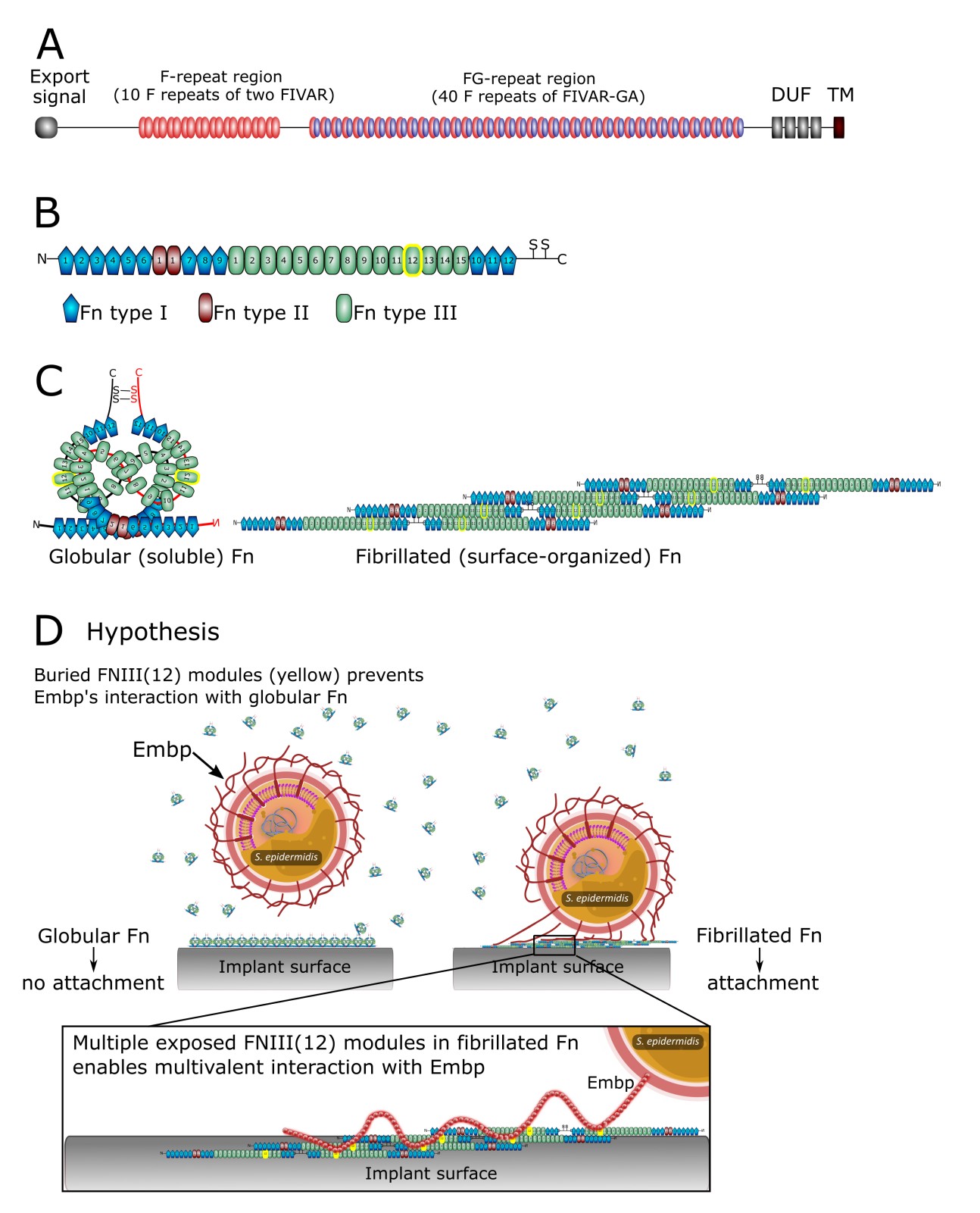

**Figure 1.** The structure of extracellular matrix-binding protein Embp (**A**), its ligand fibronectin (**B and C**), and our hypothesis of selective binding of Embp to the fibrillated form of fibronectin via a multivalent interaction (**D**). (**A**) Embp consists of 50 fibronectin (Fn)-binding repeats (10 F-repeats and 40 FG-repeats). (**B**) The structure of Fn. Embp binds to FN12 of the Fn type III (highlighted in yellow). (**C**) Representation of globular and fibrillated Fn. Fn is a dimer held together by two disulfide brides, and interactions between FN(III)11–14 on one Fn and FN(III)1–4 on the other Fn. In fibrillated Fn, these

*Figure 1 continued on next page*

*Figure 1 continued*

domains are exposed. (**D**) Hypothesis for Embp's interaction with Fn on surfaces and in solution. (**A**) has been adapted from Figure 1 in ***Büttner et al., 2020***. (**B**) has been adapted from Figure 4A in ***Büttner et al., 2020***.

interacting with adsorbed Fn when the same protein is also abundant in a soluble form in blood. Presumably, Fn-binding proteins on the bacterial cell surface become occupied with soluble Fn before being able to interact with Fn on the implant surface. The aim of this study was to determine how pathogens overcome this dilemma and bind to host proteins on tissue or implant surfaces while ignoring soluble forms of the same protein. Understanding the pathogens' ability to selectively colonize implant surfaces reveals conceptual mechanisms for how pathogens control their location and fate in the host.

In this study, we investigate Embp's interaction with Fn. Fn circulates in bodily fluids in a compact globular form (***Rocco et al., 1984***), while fibrillated Fn contributes to the assembly of the extracellular matrix of tissue (***Baneyx et al., 2001***). It is the stretching of Fn upon interacting with cell surface integrins, which exposes self-binding domains and triggers Fn fibrillation. This mechanism ensures that Fn only fibrillates in the extracellular matrix of tissue and not in the bloodstream (***Zhong et al., 1998***). We hypothesize that *S. epidermidis* interacts selectively with fibrillated Fn because FN12 is buried in the globular form of the protein. Furthermore, we hypothesize that a fibrillated ligand provides an opportunity for a multivalent interaction with the many repetitive F- and FG-repeats of the Embp (***Figure 1D***).

Using a model system of polymer-coated surfaces that facilitate Fn adsorption in either globular or fibrillated conformation, we probed Embp's interaction with Fn by flow cell and advanced atomic force microscopy (AFM) experiments. The AFM can be used as a tactile tool for biological samples, which employs a cantilever with a sharp tip scanned over a sample with low forces in air or liquid environments (***Dufrêne et al., 2017***; ***Müller and Dufrêne, 2008***). We used a mode of AFM in which the sharp tip intermittently contacts the surface while scanning over it to create a 3D real-space image. Doing so enabled us to reveal the globular and fibrillar Fn conformations on polymer surfaces. Beyond the surface morphology, AFM can be used to measure the quantitative forces acting between the tip and the sample surface (***Zhang et al., 2014***). The latter provides insights in many biological binding events by the modification of the probe, for example, by attaching a single cell at the end of the cantilever instead of a sharp tip (***Viljoen et al., 2021***). Attaching a cell or proteins on the probe and using it to measure quantitative forces on the sample of interest are referred as single-cell (SCFS) or single-molecule force spectroscopy (SMFS) which enables the observation of the nature of biomolecular binding events and their dynamics. Using native and recombinant Embp (rEmbp) in a series of analyses at the population, single-cell, and single-molecule levels, we confirmed that Embp selectively interacts with fibrillated Fn. The interaction is a Velcro-like mechanism where multiple binding domains must interact simultaneously to facilitate strong attachment. Such strong attachment via a single protein is particularly beneficial under high sheer stress, such as in the vascular system, and it was exactly under these conditions that Embp gave the cells an advantage. Embp homologs are present in other important pathogens capable of biofilm formation in the vascular system, and our study reveals a mechanism for how bacteria may accomplish this feat.

## Results

### Embp does not interact with soluble Fn

We hypothesized that Embp selectively binds to fibrillated Fn, which would allow the bacteria to colonize surfaces via Fn without being blocked by soluble Fn in the bloodstream (***Figure 1D***). To study the interaction between Embp and Fn, we expressed Embp fusion proteins comprised of either 5 F-repeats (Embp_5F) or 9 FG-repeats (Embp_9FG), each fused to the native export signal and anticipated C-terminal cell wall anchor region (***Büttner et al., 2020***), in the surrogate host *Staphylococcus carnosus* TM300, which has no other mechanisms for attachment to Fn. The full-length Embp is too large to clone into a surrogate host, and it was therefore not possible to investigate the full-length Embp protein. However, expression of the two different Embp fragments allowed us to study their interactions individually. The presence of Embp fragments on the cell surface was verified by immunofluorescence staining (***Figure 2—figure supplement 1***).

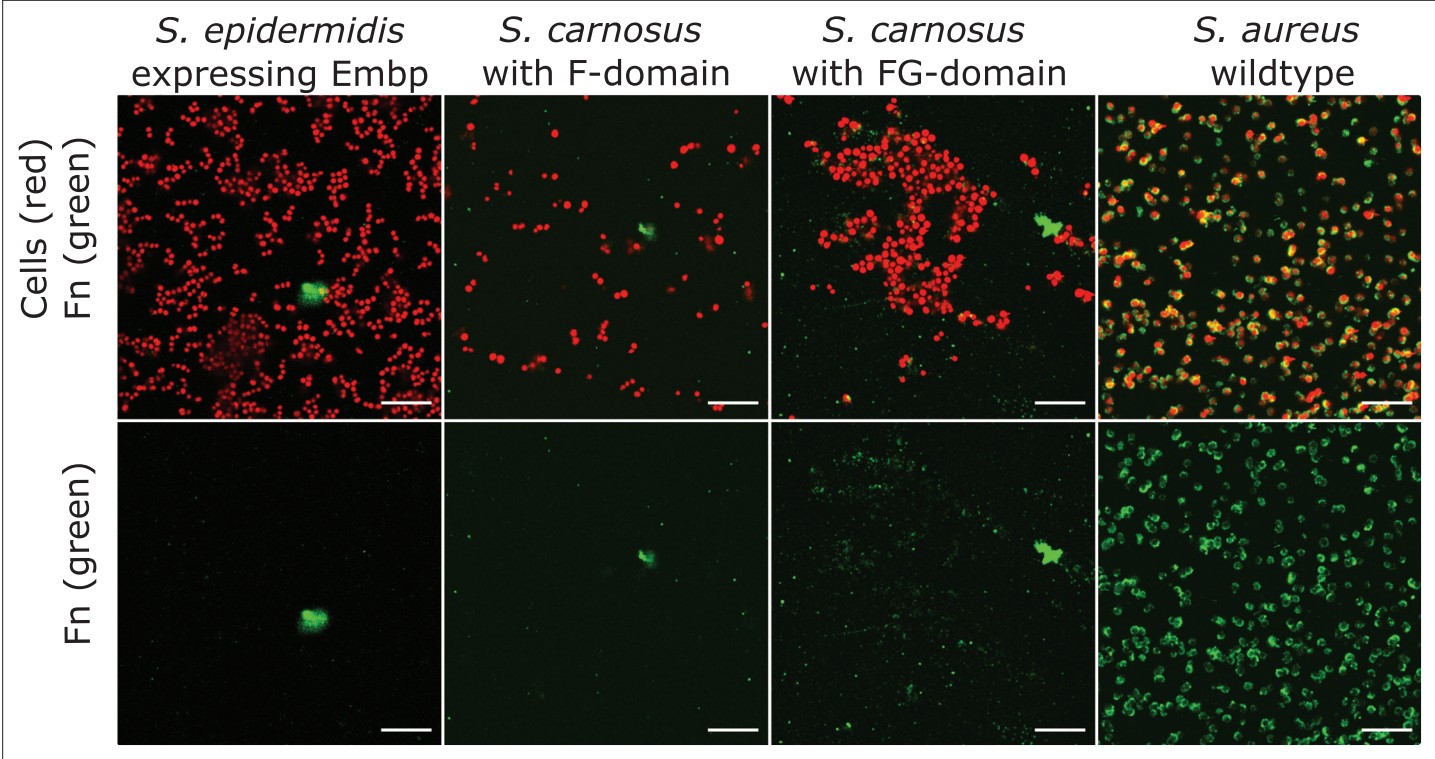

**Figure 2.** Soluble fibronectin (Fn) does not bind to extracellular matrix-binding protein (Embp). Interaction of soluble Fn with *Staphylococcus epidermidis* 1585Pxyl/tet *embp* overexpressing Embp or *Staphylococcus carnosus* TM300 expressing recombinant Embp (5F- or 9 FG-repeats) was detected by fluorescence microscopy. *Staphylococcus aureus* was used as positive control. Bacteria were stained with SYTO 9 (depicted as red), and Fn bound to the cell surface was detected by immunolabeling, using anti-Fn mouse IgG primary antibody, and anti-mouse IgG conjugated with Alexa Fluor 635 as secondary antibody (depicted as green). Top panel shows overlay of bacteria (red) and Fn (green). Bottom panel shows Fn only (green). Scale bar = 10 μm.

The online version of this article includes the following source data and figure supplement(s) for figure 2:

**Source data 1.** Zip file containing original confocal laser scanning microscopy (CLSM) images for *Figure 2*.

**Figure supplement 1.** Verification of extracellular matrix-binding protein (Embp) fractions on the surface of *Staphylococcus carnosus* expressing 5F-repeats.

**Figure supplement 1—source data 1.** Zip file containing original confocal laser scanning microscopy (CLSM) images for *Figure 2—figure supplement 1*.

Neither F- nor FG-repeats facilitated adsorption of soluble fluorescently conjugated Fn to the surface of *S. carnosus* (*Figure 2*). The native Embp expressed by *S. epidermidis* did not bind soluble Fn either (*Figure 2*), concluding that Embp does not interact with Fn in its soluble, globular conformation.

## Embp interacts exclusively with fibrillated Fn

In order to further investigate Embp's interaction with Fn in different conformations, we produced a model system in which Fn was adsorbed to a surface in either the globular or fibrillated conformation. Previous research had shown that Fn fibrillates when adsorbed on surfaces coated with poly(ethyl acrylate) (PEA), while it remains globular on poly(methyl acrylate) (PMA) (*Llopis-Hernández et al., 2016*; *Rico et al., 2009*; *Salmerón-Sánchez et al., 2011*). The two polymer coatings have similar physicochemical properties, but the ethyl side group of PEA provides sufficient mobility of the adsorbed protein to facilitate fibrillation (*Bieniek et al., 2019*; *Guerra et al., 2010*). The presence of polymer coatings was confirmed by AFM (*Figure 3—figure supplement 1*) and X-ray photoelectron spectroscopy (XPS) (*Figure 3—figure supplements 2 and 3*).

Upon adsorption to the polymer coating, Fn spontaneously organized into a fibrillated network on PEA while remaining globular on PMA (*Figure 3A and B*). In order to ascribe any differences in adhesion to the conformation and not the amount of Fn, we analyzed the quantity of protein on

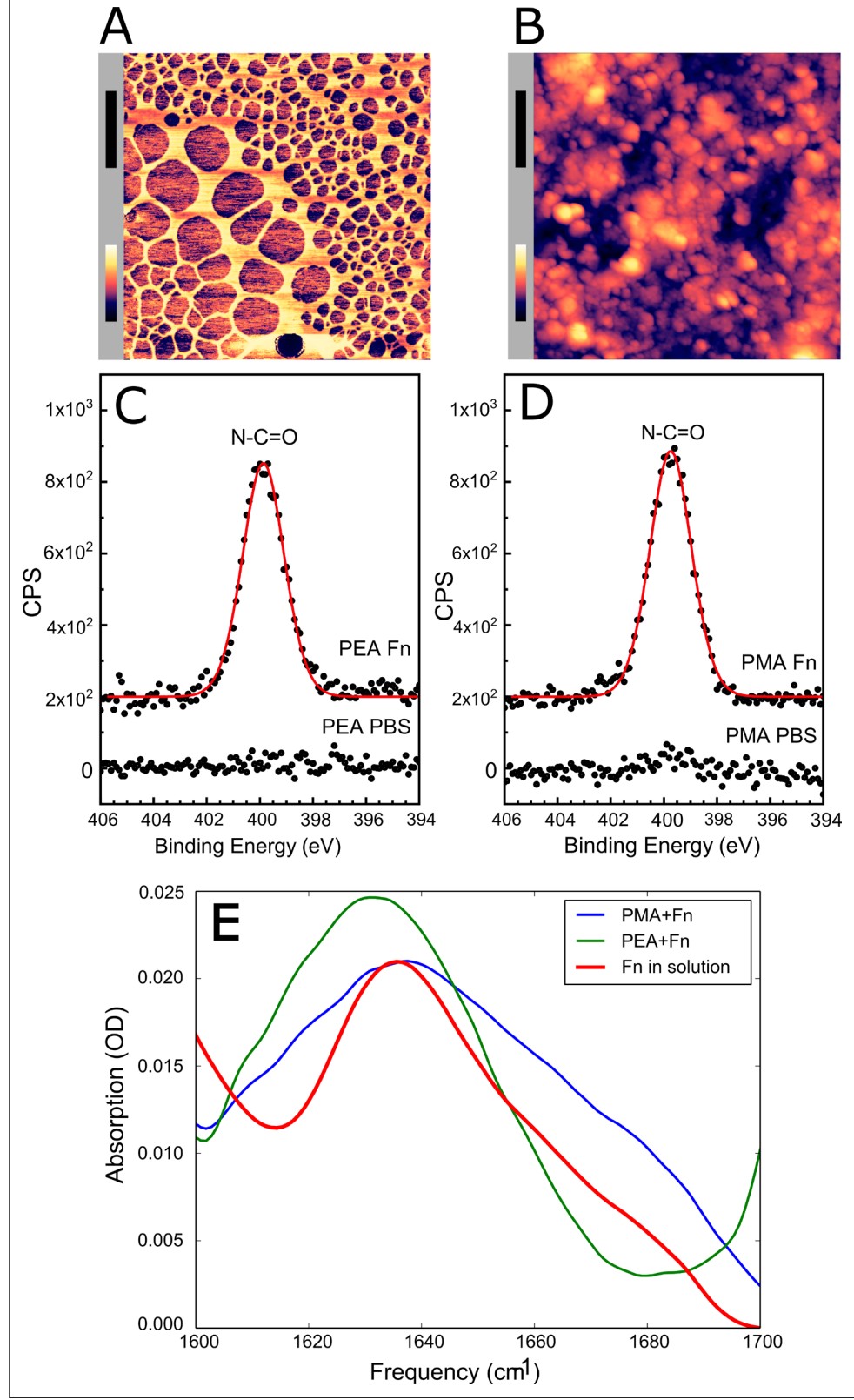

**Figure 3.** Adsorbed fibronectin (Fn) remains globular on poly(methyl acrylate) (PMA) and fibrillates on poly(ethyl acrylate) (PEA)-coated surfaces. Atomic force microscopy (AFM) imaging shows the structure of adsorbed Fn on (**A**) PEA and (**B**) PMA (xy-scale bar (black)=500 nm, height scale bar (color)=115 nm). X-ray photoelectron spectroscopy (XPS) analysis of the samples shows similar chemical composition of Fn adsorbed to (**C**) PEA and (**D**) PMA,

*Figure 3 continued on next page*

*Figure 3 continued*

indicating that two polymer surfaces are both covered by Fn. (**E**) Fourier transform infrared (FTIR) spectral shape and intensity confirms that Fn adsorbed to PMA is similar to Fn in solution.

The online version of this article includes the following source data and figure supplement(s) for figure 3:

**Source data 1.** Zip file containing original and processed atomic force microscopy (AFM) images for *Figure 3A and B*.

**Figure supplement 1.** Surface topography of poly(methyl acrylate) (PMA)- and poly(ethyl acrylate) (PEA)-coated glass.

**Figure supplement 1—source data 1.** Zip file containing original and processed atomic force microscopy (AFM) images.

**Figure supplement 2.** Verification of polymer coating and adsorbed fibronectin (Fn) by X-ray photoelectron spectroscopy (XPS).

**Figure supplement 3.** High-resolution $C_{1s}$ X-ray photoelectron spectroscopy (XPS) plots of peaks characteristic of protein addition on polymer surfaces.

**Figure supplement 4.** Various variants of processing of the Fourier transform infrared (FTIR) spectra, indicating the relative contributions of scattering, protein absorption, and polymer absorption.

**Figure supplement 5.** Results of least-squares fit before (top) and after (bottom) fibronectin (Fn) incubation of the spectra of the poly(ethyl acrylate) (PEA)- to the poly(methyl acrylate) (PMA)-coated surfaces, in order to obtain the relative number of PEA and PMA molecules on the surfaces.

the two surfaces. XPS analysis determined that the amount of adsorbed protein was similar on the two polymer surfaces (*Figure 3C and D*, and *Table 1*). The XPS survey scan and high-resolution $C_{1s}$ XPS plots are shown in *Figure 3—figure supplements 2 and 3*. The conformational differences of adsorbed Fn on the two coatings was corroborated by Fourier transform infrared (FTIR) spectra, in which the peak positions indicate that Fn adsorbed to PMA adopts a mostly antiparallel β-sheet type secondary structure (*Barth and Zscherp, 2002*), similar to the globular, solution-state spectrum, while Fn on PEA adopts a more extended parallel β-sheet type structure (*Figure 3E*, *Figure 3—figure supplements 4 and 5*).

After validating the model system, Embp-mediated bacterial attachment to fibrillated and globular Fn was measured using a flow cell system where the number of attached bacteria was counted by microscopy. Very few bacteria attached to the polymer coatings in the absence of Fn, and only fibrillated Fn stimulated attachment of *S. carnosus* expressing Embp_5F or Embp_9FG (*Figure 4A*). Fn consists of two nearly identical subunits linked by a pair of disulfide bonds at the C terminal (*Kar et al., 1993*). Each subunit consists of three domains: F1, F2, and F3 (*Potts and Campbell, 1996*). The globular and compact conformation of Fn is sustained by intramolecular electrostatic interactions between F1 1st-5th, F3 2nd-3rd, and F3 12th-14th repeat (*Johnson et al., 1999*; *Maurer et al., 2015*).

**Table 1.** X-ray photoelectron spectroscopy (XPS) survey spectrum atomic percent compositions for polymer poly(ethyl acrylate) (PEA) and poly(methyl acrylate) (PMA) and either phosphate buffered saline (PBS) or PBS with fibronectin (Fn).

| Element | PEA PBS [%] | PEA Fn [%] | PMA PBS [%] | PMA Fn [%] |
|---|---|---|---|---|
| $C_{1s}$ | 79.5 (1.0) | 55.8 (1.0) | 21.4 (1.4) | 67.4 (3.8) |
| $O_{1s}$ | 19.6 (1.0) | 27.0 (1.1) | 55.5 (1.2) | 21.4 (1.3) |
| $N_{1s}$ | n.d. | 8.3 (0.4) | n.d. | 7.7 (1.5) |
| $Si_{2p}$ | 0.9 (0.4) | 0.1 (0.1) | 19.8 (0.8) | 0.2 (0.2) |
| $Na_{1s}$ | n.d. | 3.6 (0.1) | 3.2 (0.7) | 1.9 (0.6) |
| $Cl_{2p}$ | n.d. | 2.5 (0.3) | 0.1 (0.1) | 1.4 (0.3) |
| $P_{2p}$ | n.d. | 2.7 (0.1) | n.d. | n.d. |
| $K_{2p}$ | n.d. | n.d. | n.d. | n.d. |

Note: not detectable (n.d.); (.)=standard deviation; [%]=atomic percent (sample mean, *n*=3).

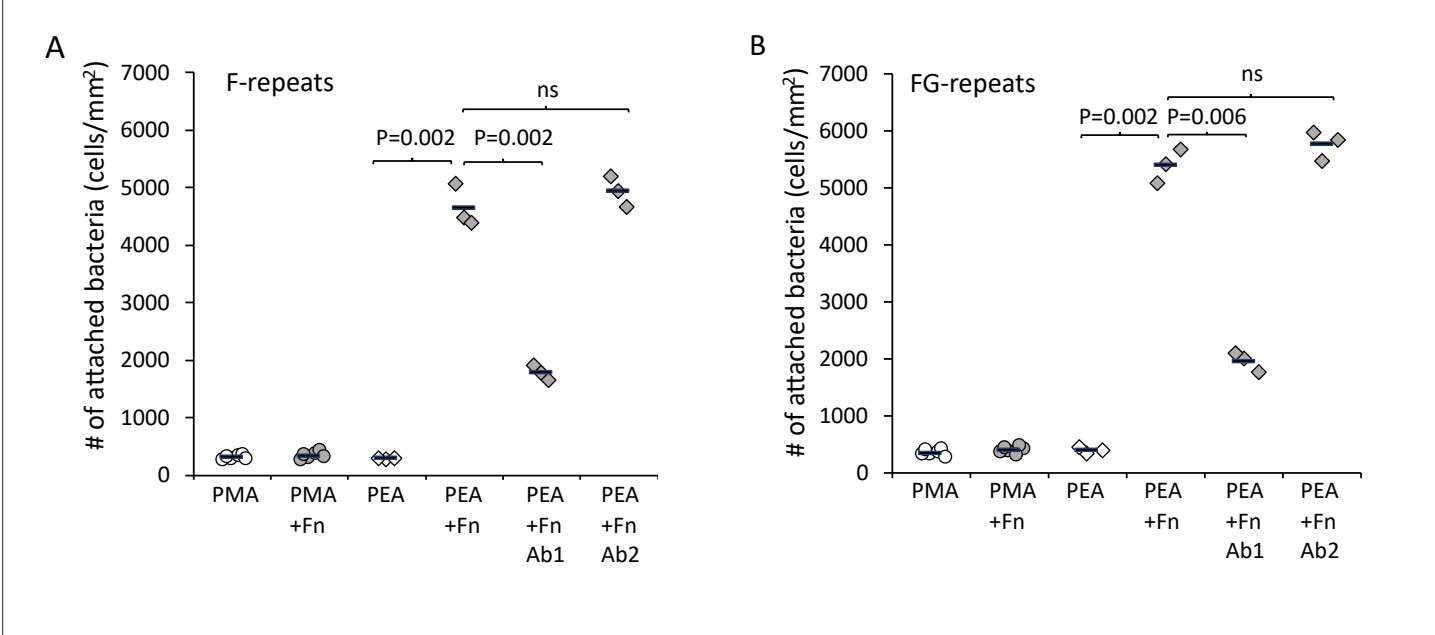

**Figure 4.** Extracellular matrix-binding protein (Embp) only mediates bacterial attachment to fibrillated fibronectin (Fn). *Staphylococcus carnosus* TM300 expressing either F- (panel **A**) or FG-repeats (panel **B**) were passed through flow cells for 2 hr before enumeration of attached cells by microscopy. Adsorption of Fn only promoted attachment on poly(ethyl acrylate) (PEA)-coated surfaces where Fn fibrillated. Blocking of Fn at FnIII 12th-14th by specific antibodies (Antibody 1) resulted in decreased bacterial attachment, which indicates that Embp binds to this domain. Blocking of Fn at a different domain (FnIII 5th, Antibody 2) was included as a control for non-specific blocking of Fn by the antibodies. This antibody had no effect on bacterial attachment. p-Values are indicated (two-tailed t-test).

The online version of this article includes the following source data for figure 4:

**Source data 1.** Excel file with raw data and statistical calculations for the number of attached bacteria quantified by brightfield microscopy.

Binding sites in these regions remain buried in the globular conformation; however, upon fibrillation on a surface or tissue interface, these binding sites become exposed (*Klotzsch et al., 2009*). Since Embp only binds to fibrillated Fn, we hypothesize that it interacts with epitopes that are buried in the globular conformation, but become exposed when Fn fibrillates. Indeed, it was previously reported that *S. epidermidis* binds near the C terminal of Fn (*Arciola et al., 2003*), and studies of recombinant Fn verified the interaction between Embp and the 12th repeat of the F3 domain (*Christner et al., 2010*). This repeat may be one of several interaction points and has not been confirmed in full-length Fn adsorbed in its natural conformation. To test the interaction between the 12th repeat of the F3 domain and the Fn-binding F- and FG-repeats, we repeated cell adhesion analysis on Fn-coated PEA after blocking the C-terminal heparin-binding domain II (F3 12th-14th repeat) with antibody sc-18827. Control samples were blocked with antibody F0916 specific for the F3 5th repeat (*Figure 4*). Blocking the F3 12th-14th repeat decreased the adherence of *S. carnosus* by approximately 62% for Embp_5F and 64% for Embp_9FG (*Figure 4*), supporting that Embp interacts with this subdomain. As the adherence was not completely abolished by blocking the Fn-binding site, we cannot exclude the possibility that Embp interacts with other epitopes in Fn. However, the F3 12th-14th repeat is of major significance.

## F and FG modules attach to Fn

After learning that Embp interacts exclusively with fibrillated Fn, we probed the strength of this interaction by single-cell atomic force spectroscopy. Single *S. carnosus* expressing Embp_5F or Embp_9FG were attached to colloidal AFM probes, approached to an Fn-coated PMA or PEA surface with controlled force, and then retracted to detect the force needed to detach the cell from the surface. As expected, the force-distance curves obtained from these experiments show that both F and FG fragments bind to fibrillated but not to globular Fn. The average maximum adhesion force between *S. carnosus* and surfaces with fibrillated Fn was 1.19±0.21 and 1.16±0.18 nN, respectively, for *S.*

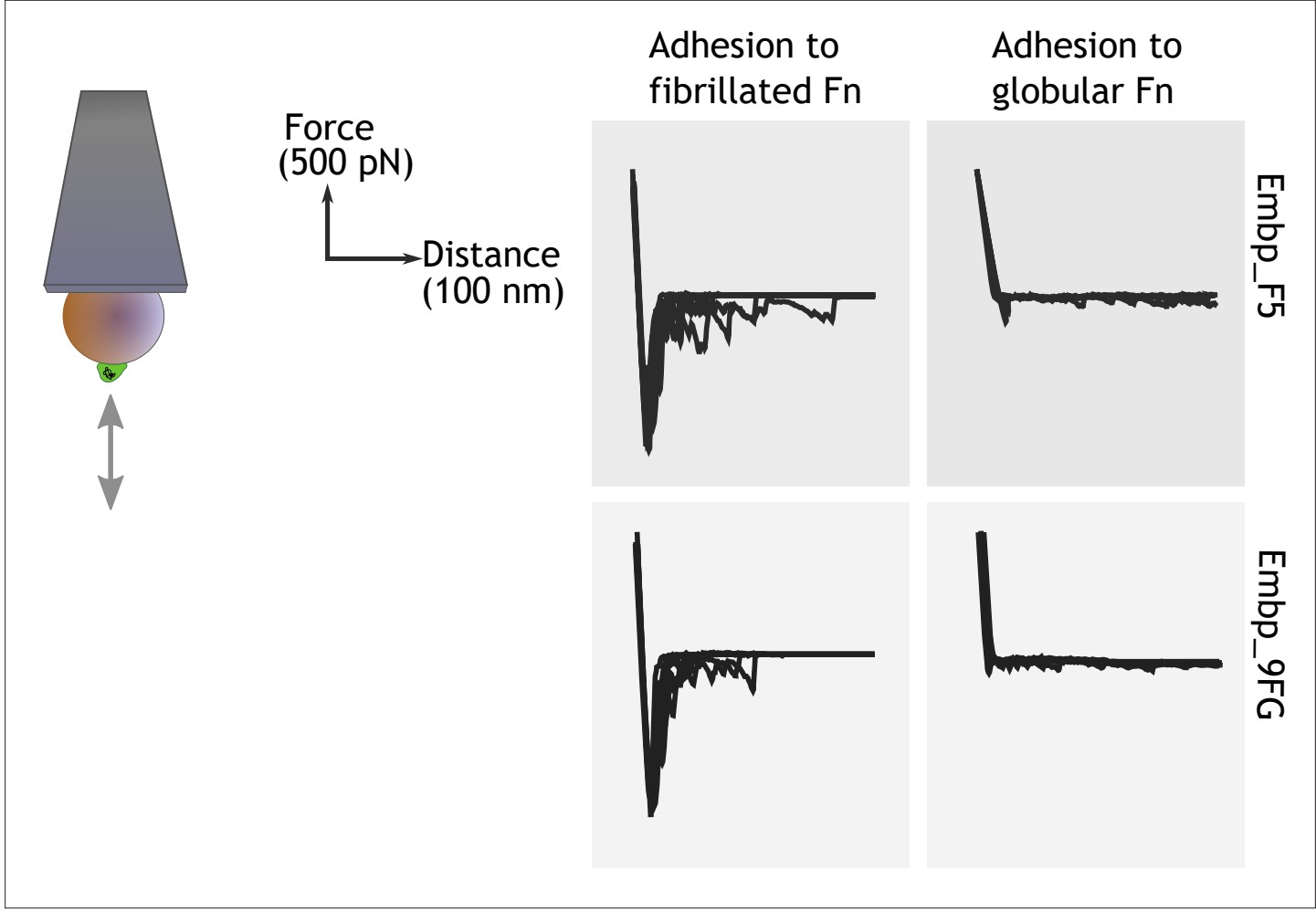

**Figure 5.** Single-cell force spectroscopy shows that both the F- and FG-repeats adheres strongly to fibrillated fibronectin (Fn). Single *Staphylococcus carnosus* cells expressing either F- or FG-repeats were immobilized on a colloidal atomic force microscopy (AFM) cantilever, and force-distance curves were measured by approaching and retracting the cantilever to surfaces with fibrillated or globular Fn. Adhesion events are recognized as negative peaks on the force axis below the horizontal baseline.

The online version of this article includes the following source data for figure 5:

**Source data 1.** Zip file containing raw force spectroscopy data and screenshots to document system setup.

*carnosus* expressing Embp_5F or Embp_9FG (*Figure 5*). In contrast, the corresponding adhesion force to surfaces with globular Fn was only 0.16±0.09 and 0.12±0.04 nN. The adhesion force and the shape of the force-distance curves reflect multiple binding events between the cell and the Fn-coated surface. The multiple binding events could either be due to multiple Embp fragments on the cell surface interacting with Fn or multiple interactions between a single Embp fragment and Fn.

### Embp binds to fibrillated Fn in a Velcro-like manner

Embp contains 50 Fn-binding repeats, and it must be costly for *S. epidermidis* to produce this enormous 1 MDa protein. How might *S. epidermidis* benefit from the many repetitive binding domains? We hypothesize that multivalent interactions can occur if the ligand for this giant adhesin is fibrillated, resulting in presentation of multiple binding domains in close proximity. Such multivalent binding would work like Velcro, as many weak binding events result in strong attachment. Such a Velcro effect could provide adhesion forces strong enough to attach *S. epidermidis* to Fn via a single Embp protein. To investigate this hypothesis, we expressed and purified rEmbp fragments that contained 1, 4, and 15 repeats of FG-repeats, attached them to an AFM cantilever using 6× His-NTA interaction, and quantified their interaction with Fn by SMFS. In agreement with previous experiments, the FG-repeat

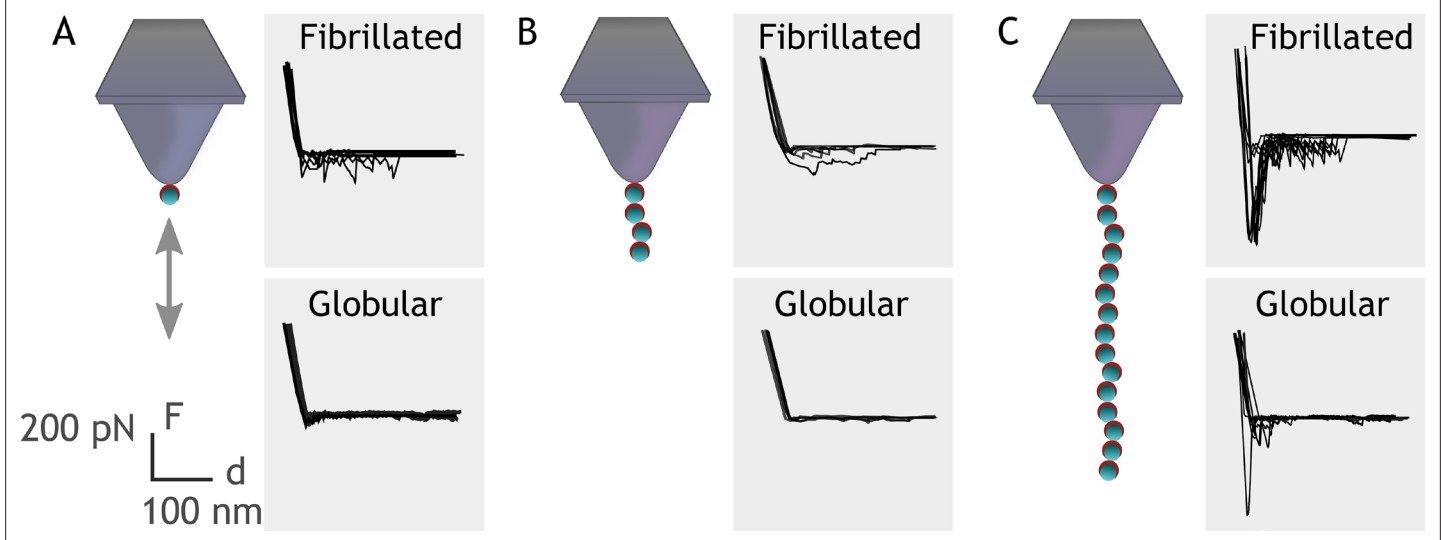

**Figure 6.** Single-molecule force spectroscopy shows that multiple FG-repeats are needed to detect binding to fibronectin (Fn). Recombinant extracellular matrix-binding protein (Embp) consisting of either 1 (**A**), 4 (**B**), or 15 (**C**) FG-repeats was tethered to a chemically modified silicon probe trough 6× His-NTA interaction. Force-distance curves were measured toward fibrillated Fn on poly(ethyl acrylate) (PEA) and globular Fn on poly(methyl acrylate) (PMA).

The online version of this article includes the following source data for figure 6:

**Source data 1.** Zip file containing raw force spectroscopy data and screenshots to document system setup.

did not interact with the globular form of Fn (*Figure 6*). The interaction force of a 1 or 4 FG-repeat with fibrillated Fn was also insufficient to be detected. However, the interaction force of 15 FG-repeats was 432±48 pN with fibrillated Fn, confirming the value of multidomain interaction with the fibrillated ligand (*Figure 6*).

## Embp is necessary for attachment under high flow

In our investigating of the interaction mechanism between Embp and Fn, we used fusion proteins that contained only a few of the F- or FG-repeats displayed on the surface of *S. carnosus* which has no other adhesive proteins. However, *S. epidermidis* has many other cell wall anchored adhesins, and the key to understanding Embp's role in *S. epidermidis*' pathogenicity therefore lies in understanding the circumstances under which Embp-producing *S. epidermidis* strains have an advantage. If oriented perpendicular to the cell surface, Embp could potentially stretch several hundred nm from the cell surface. We measured the hydrodynamic radius of *S. epidermidis* overexpressing Embp, and confirmed that it was significantly larger than for *S. epidermidis* lacking Embp (2.3±0.4 µm vs. 1.3±0.2 µm, two-tailed t-test, $n$=3, p=0.02).

We speculated that Embp would be more effective than other adhesins when *S. epidermidis* is attaching to Fn under high sheer stress. We therefore compared attachment of the two strains at low flow (0.03 dyn cm$^{-2}$) and high flow (0.53 dyn cm$^{-2}$). At low flow, Embp increased attachment to Fn by approximately 20% (two-tailed t-test, p=0.03), and at high flow, attachment was only possible Embp was expressed (*Figure 7*).

## Discussion

In this study, we show that the giant cell surface protein Embp exclusively binds to fibrillated Fn because the binding site located at F3 12th –14th repeat is not accessible in the globular, soluble form of Fn. This discovery has implications for our understanding of how *S. epidermidis* colonizes host tissue and biomedical implants. Colonization and biofilm formation is the only virulence factor of *S. epidermidis*, and it is therefore imperative to cause disease (*Both et al., 2021*).

Implants provide a surface for attachment and are therefore vulnerable to biofilm infections. The implant surface is immediately covered by host proteins when it is inserted into the body, and

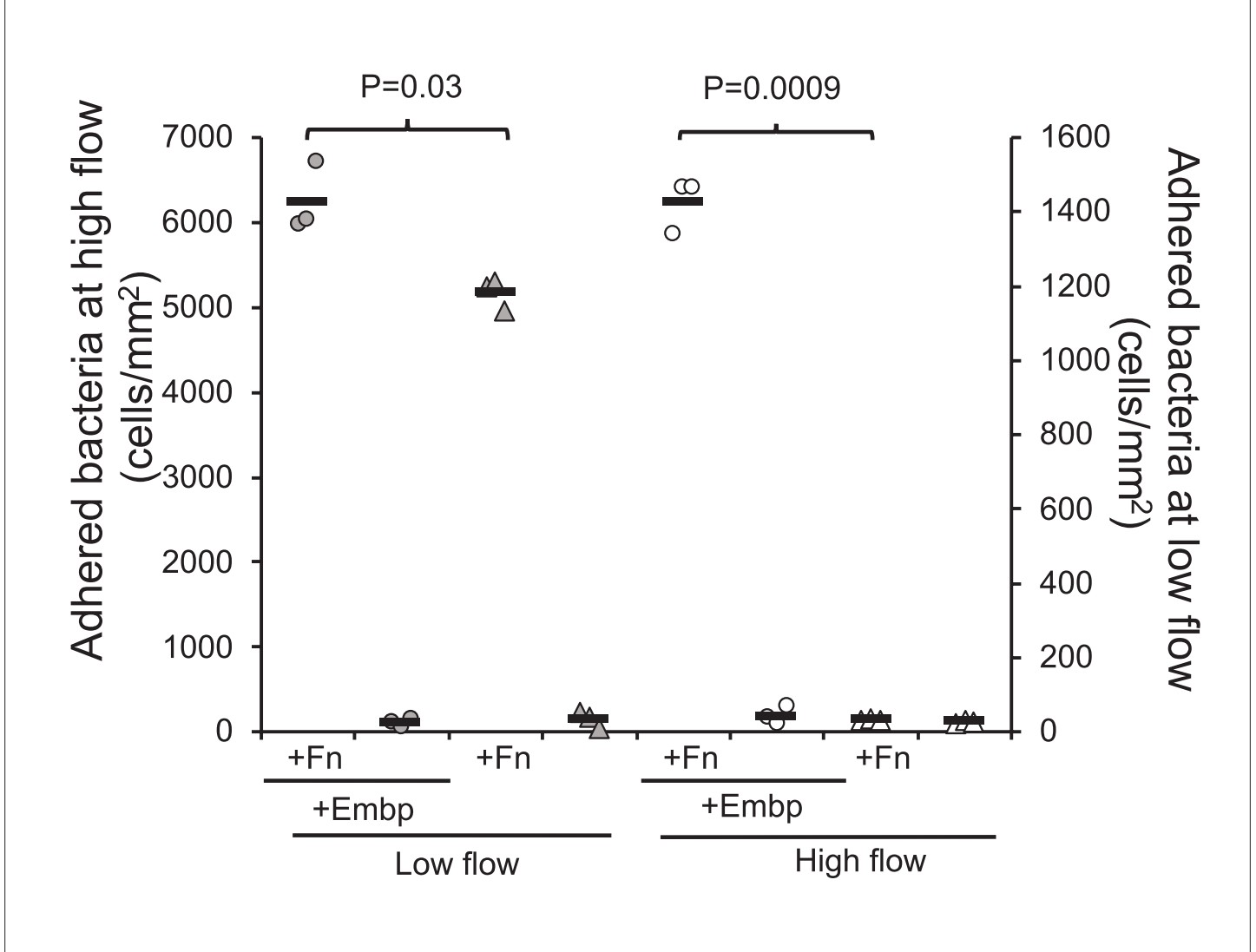

**Figure 7.** Extracellular matrix-binding protein (Embp) is essential for attachment under high flow. Attachment to poly(ethyl acrylate) (PEA) +/-Fn was quantified for *Staphylococcus epidermidis* overexpressing (circles) or lacking (triangles) Embp at high flow (gray symbols, left y axis) and low flow (white symbols, right y axis). Expression of Embp led to a 20% increase in attachment at low flow (two-tailed t-test, unequal variance, p=0.03). At high flow, only the Embp-expressing strain could attach (two-tailed t-test, p=0,0009). Black bars show sample mean.

The online version of this article includes the following source data for figure 7:

**Source data 1.** Excel file with raw data, calculations, and statistical analysis for quantification of bacterial attachment, shear force calculation, and measurement of hydrodynamic radius.

bacterial attachment is assumed to occur via specific receptor-ligand interactions between adhesins on the bacterial cell and host proteins adsorbed to the implant surface. The abundance of the same host proteins in solution poses a dilemma: How can bacteria attach to implant surfaces via proteins that are also available in solution? Based on previous research (*Büttner et al., 2020*; *Christner et al., 2010*), we hypothesized that adsorption-induced conformational changes can affect how accessible a host protein is for bacterial adhesins. In the case of Embp and Fn, the fibrillation of Fn on the implant surface is decisive for its availability as a ligand for bacterial attachment. The selective interaction with fibrillated Fn illustrates how biology has solved the need for interaction with a protein in one location (the extracellular matrix of host tissue) while ignoring the same protein in another location (the bloodstream).

Our results also challenge notion that adsorbed host proteins always assist bacterial colonization. Motivated by this assumption, much research has been devoted to developing materials or

coatings that prevent protein adsorption altogether. However, our results suggest that biomaterials' susceptibility to bacterial colonization does not only depend on protein adsorption, but also on the conformation of adsorbed proteins – a concept that is familiar to cell biologists studying adhesion of mammalian cells, but less so to microbiologists studying bacterial attachment. The role of host protein conformation in bacterial attachment could explain the conflicting results from in vitro studies of how serum proteins affect attachment of staphylococci. Some studies report increased attachment (*Christner et al., 2010*; *Paharik and Horswill, 2016*; *Williams et al., 2002*; *Maxe et al., 1986*), while others report the opposite (*Paulsson et al., 1994*; *Pestrak et al., 2020*). Perhaps these discrepancies reflect differences in how the underlying material affected the conformation of adsorbed serum proteins. Our study investigated bacterial attachment in a simplified model system with only one host protein, but if the concepts hold true in serum, it opens a door to controlling bacterial attachment by manipulating the conformation of adsorbed proteins, which is a more manageable task than avoiding protein adsorption entirely.

The large number of repetitive domains in Embp and its homologs in *S. aureus* (Ebh) and *Streptococcus defectivus* (Emb) is unusual among bacterial adhesins, and we wondered how pathogens benefit from producing such large proteins. One benefit could be that the adhesin protrudes from the cell surface, which makes it easier to overcome electrostatic repulsion and reach its ligand. At present, no experimental evidence is available to directly demonstrate the organization of Embp on the cell surface. However, given the stretched overall architecture revealed by structural analysis (*Büttner et al., 2020*), it appears plausible the molecule's length is similar to that of Ebh in *S. aureus*. Ebh is a 1.1 MDa protein with 52 repetitive domains, and its length was predicted to be 320 nm (*Tanaka et al., 2008*). We show that the hydrodynamic radius of *S. epidermidis* overexpressing Embp was approximately 1 µm larger than *S. epidermidis* lacking Embp. It is thus likely that Embp extends well beyond the electric double layer.

Another advantage of adhesins with repetitive binding domains is the possibility for multivalent interaction with the ligand. Such multivalency is only possible if the ligand is fibrillated and thereby presenting many copies of its binding domain in proximity of one another. Indeed, we showed that a single FG module interacts weekly with Fn, while the interaction force of 15 FG-repeats was 432±48 pN. We therefore propose that the selective interaction with fibrillated Fn is caused by (i) the exposure of an otherwise buried binding domain and (ii) the possibility of a stronger, multivalent interaction between multiple FG-repeats and the fibrillated ligand.

The typical strength of receptor-ligand bonds is about 20–200 pN (*Müller et al., 2009*). Hence, the interaction between 15 FG-repeats and Fn resulted in a very strong bond, and the interaction with full sized Embp is likely much stronger. Moreover, the force required to detach Embp from fibrillated Fn will depend on whether all binding repeats detach at once, or whether detachment can occur from the serial unbinding of the repeats one by one. This is akin to detachment of Velcro: The attachment is very strong, but detachment can be obtained by the serial unbinding of individual interactions. The force-distance curves can provide information about how detachment occurs. On a close inspection, it is clear that initial retraction events are the strongest and provide the highest adhesion force, indicating that multiple bonds are broken at the same time (*Figure 6B*). But we also observe multiple subsequent perturbations as evidence to serial rupture, unfolding or stretching events. This holds true for interactions with 15 FG-repeats, but multiple perturbations are also observed for a single FG-repeat. In this case, the perturbations likely arise from the stretching and ruptures of surface-bound Fn yielding an average adhesion force of 92±27 pN, which is around the detection limit for this setup. Similar results were observed for 4 repeating units with average adhesion force of 68±43 pN. On surfaces with globular Fn, little to no interaction was observed, although 15 repeating units undergo certain irregular interactions. An explanation for the observed irregularities could be simply the protein's self-occupation/entanglement as would often happen with long polymer chains. Additionally, as can be seen from *Figure 3*, the surface distribution of Fn can vary, depending on the local density. This may be an insignificant problem for the dynamics of cell populations, however, when it comes to assessing the behavior of an individual cell or protein, the local density can affect the results.

In conclusion, the specific binding domain and repetitive structure of Embp provides an opportunity for *S. epidermidis* to interact selectively and strongly with fibrillated Fn, using a single adhesive protein. The open question is how this unique Velcro-like of interaction plays into the pathology of *S. epidermidis* and other pathogens that contain homologs of this protein in their genome. The genome

of *S. epidermidis* is highly variable, and not all isolates possess the same repertoire of genes for host colonization and biofilm formation (*Post et al., 2017*). Embp, however, is present in two-thirds of *S. epidermidis* isolates from orthopedic device-related infections (*Post et al., 2017*) and 90% of isolates from bloodstream infections (*Salgueiro et al., 2017*), which indicates some importance for its pathogenicity. In a recent study of adhesive vs. invasive *S. epidermidis* isolates, Embp was not more frequent in any of the groups (*Both et al., 2021*). Hence, there is no indication of a general role of Embp for invasiveness vs. colonization. We speculate that strong attachment via a single protein like Embp could be particularly advantageous in locations where shear forces make attachment difficult. Biofilms generally do not form in blood vessels unless there is an implant or a lesion on the endothelium. Infections like endocarditis often start with such a lesion (*Steckelberg and Wilson, 1993*), which makes the site susceptible to bacterial attachment. Fibrillated Fn forms on the surface of platelets in the early stages of wound healing (*To and Midwood, 2011*), and perhaps the abundance of fibrillated Fn plays role in the elevated infection risk by *S. epidermidis* strains that carry Embp. The abundance of Embp among isolates from such infections remains to be investigated.

We show here that Embp makes a moderate contribution to attachment under low flow, leading to a 21% increase in the number of attached bacterial, while it is essential for attachment under high flow (*Figure 7*). Other than Embp, only the autolysin Aae has been reported to bind Fn based on in vitro studies of purified proteins (*Heilmann et al., 2003*), and *S. epidermidis* 1585 does contain this gene. It is possible that Aae and even non-specific interaction forces contributed to adhesion of *S. epidermidis* at low flow, whereas only Embp attaches with sufficient force to prevent detachment at high flow.

Previously published studies have shown that the binding force of some bacteria adhesins is promoted by external mechanical forces, such as shear stress when under flow, achieved in the 'dock, lock, and latch' mechanism observed for the Clf-Sdr family of adhesins when binding to fibrinogen (*Herman et al., 2014*; *Herman-Bausier et al., 2017*; *Vanzieleghem et al., 2015*; *Viela, 2019*). However, our data do not suggest that increased shear stress would strengthen the interaction force between Embp and Fn, as the overall attachment rate of bacteria decreases by approximately 80% at the high flow rate. Furthermore, the structural analysis of Embp carried out by *Büttner et al., 2020*, points out that the Embp represents a novel mode of interaction between bacterial adhesins and host extracellular matrix proteins. Considering the giant size of Embp, one can speculate that stretching of Embp due to lateral external mechanical forces will increase its contact area with surface-organized Fn, and thereby increase the probability of successful binding events leading to adhesion. However, our data do not reveal whether this is the case. What we can conclude with certainty is that Embp is essential for attachment when host colonization is challenged by shear forces, and this observation suggests that Embp may be significant for biofilm formation in the cardiovascular system, for example, on cardiovascular grafts. Future research in animal models will determine Embp's role in colonization of endothelial cells and implants in the cardiovascular system.

# Materials and methods

## Key resources table

| Reagent type (species) or resource | Designation | Source or reference | Identifiers | Additional information |
|---|---|---|---|---|
| Strain, Strain background *Staphylococcus epidermidis* | 1585 WT | PMID:15752207 | ATCC12228 | Clinical isolate |
| Genetic reagent *Staphylococcus epidermidis* | 1585Pxyl/tet *embp* | PMID:19943904 | | Expresses Embp from inducible promotor |
| Genetic reagent *Staphylococcus epidermidis* | Δ*embp* | PMID:33082256 | | Deficient of Embp |
| Genetic reagent *Staphylococcus carnosus* | TM300 × pEmbp_5F | PMID:33082256 | | Recombinant Embp fragment (5 F-repeats) expressed in surrogate host |
| Genetic reagent *Staphylococcus carnosus* | TM300 × pEmbp_9FG | PMID:33082256 | | Recombinant Embp fragment (9 FG-repeats) expressed in surrogate host |
| Peptide, recombinant protein | Fibronectin | Sigma-Aldrich | Catalog # F0895 | 0.1% solution isolated from human plasma |
| Antibody | Anti-Embp2588 IgG (rabbit polyclonal) | Rhode lab PMID:19943904 | | (1:200) |

*Continued on next page*

*Continued*

| Reagent type (species) or resource | Designation | Source or reference | Identifiers | Additional information |
|---|---|---|---|---|
| Antibody | Anti-rabbit IgG conjugated with Alexa Fluor 635 (goat polyclonal) | Invitrogen | Catalog # A31577 | (1:300) |
| Antibody | Anti-human Fn IgG (FNIII 12–14) (mouse monoclonal) | Santa Cruz Biotech | Catalog # sc-18827 | (1:100) |
| Antibody | Anti-human Fn IgG (FNIII 5th) (Mouse monoclonal) | Sigma-Aldrich | Catalog # F0916 | (1:100) |
| Antibody | Anti-mouse IgG conjugated with Alexa Fluor 635 (goat polyclonal) | Invitrogen | Catalog # A31574 | (1:300) |
| Chemical compound, drug | Goat serum | Invitrogen | Catalog # 31873 | Blocking buffer |
| Chemical compound, drug | Ethyl acrylate | Sigma-Aldrich | Catalog # E9706-1L | Monomer for generating PEA |
| Chemical compound, drug | Methyl acrylate | Sigma-Aldrich | Catalog # M27301-1L | Monomer for generating PMA |
| Chemical compound, drug | Benzoin | Sigma-Aldrich | Catalog # B8633 | Initiator for the polymerization |
| Commercial assay, kit | Phusion High-Fidelity PCR Kit | New England Biolabs | Catalog # E0553S | |
| Commercial assay, kit | GenElute PCR Clean-Up Kit | Sigma-Aldrich | Catalog # NA1020 | |
| Commercial assay, kit | Gibson assembly ligation master mix | New England Biolabs | Catalog # E5510S | |
| Commercial assay, kit | GeneJET Plasmid Miniprep Kit | Thermo Scientific | Catalog # K0702 | |
| Other | SYTO-9 | Invitrogen | Catalog # S34854 | DNA-binding fluorescent stain |

## Bacterial strains

*S. epidermidis* 1585 WT is a clinical isolate obtained from Rhode lab in UKE, Hamburg. *S. epidermidis* 1585Pxyl/tet *embp* expresses Embp from an inducible promotor and was used for to ensure that Embp was expressed because the regulation of the native promotor is not well described. *S. epidermidis* Δ*embp* is a null mutant lacking Embp. Recombinant expression of F- or FG-repeats in the surrogate host *S. carnosus* was used to study the interaction of Embp modules without the interference from other adhesins. *S. carnosus* TM300 × pEmbp_5F expresses rEmbp consisting of 5 F-repeats, and *S. carnosus* TM300 × pEmbp_9FG expresses rEmbp consisting of 9 FG-repeats. All strains were generated previously in the Rhode lab (*Büttner et al., 2020*).

## Immunofluorescence of Embp fusion protein

Expression of Embp fusion protein in a non-adhesive surrogate host was critical for studying the interaction of Embp without interfering interactions from other adhesive proteins on the surface of *S. epidermidis*. We therefore started out by confirming the presence of Embp fragments on the surface of the surrogate host. *S. carnosus* TM300 WT, and *S. carnosus* TM300 × pEmbp_5F were grown overnight in brain heart infusion (BHI) broth with 10 µg ml$^{-1}$ chloramphenicol (Sigma-Aldrich, Germany). Expression of Embp fragments in the mutant strains was induced with 200 ng ml$^{-1}$ anhydrotetracycline (AHT) after diluting the culture 100 times in BHI. Cells were grown for 6 hr at 37°C in a shaking incubator with 180 rpm until the 600 nm optical density (OD$_{600}$) reached approximately 1. Cells were harvested by centrifugation (4000× *g* or 10 min) and resuspended in phosphate buffered saline (PBS). A droplet of the resuspended cells was placed on a SuperFrost Ultra Plus slides (Invitrogen, Waltham, MA) for 45 min to allow the bacteria to adsorb. After washing off unbound cells, the bacteria were fixed with 4% paraformaldehyde for 30 min at room temperature and washed twice with PBS. Samples were blocked with 5% goat serum (Invitrogen, Waltham, MA) for 45 min, washed, incubated with anti-Embp2588 IgG antibodies (*Christner et al., 2010*) diluted 1:200 in blocking buffer at room temperature for 1 hr, washed three times, and then incubated with the secondary antibody (anti-rabbit IgG

conjugated with Alexa Fluor 635, Invitrogen, Waltham, MA) diluted 1:300 in blocking buffer for 1 hr at room temperature. Cells were then washed three times and stained with 10 µM SYTO 9 (Invitrogen, Waltham, MA) in PBS for 10 min, washed three times, and visualized by confocal laser scanning microscopy (CLSM) (LSM700, Zeiss, Germany) using 488 excitation for SYTO 9, and 639 nm excitation Alexa Fluor 635 conjugated antibody, and a 63× Plan-Apochromat N/A 1.4 objective.

## Interaction of Embp with soluble Fn

We first investigated if *S. epidermidis* or a surrogate host expressing Embp fragments interacted with soluble Fn in its globular conformation. *S. aureus* 29213 WT (positive control for soluble Fn binding) *S. epidermidis* 1585 WT, *S. epidermidis* 1585Δ*embp* (Embp knockout), were grown in BHI without antibiotics. *S. epidermidis* 1585Pxyl/tet *embp* (Embp overexpressed) was grown in BHI with 5 µg ml⁻¹ erythromycin, *S. carnosus* TM300 × pEmbp_5F (5 F-repeats) and *S. carnosus* TM300 × pEmbp_9FG (9 FG-repeats) were grown in BHI with 10 µg ml⁻¹ chloramphenicol. Expression of Embp, F- and FG-repeats in the mutant strains was induced with 200 ng ml⁻¹ AHT after diluting the culture 100 times in BHI. Cells were grown for 6 hr at 37°C in a shaking incubator with 180 rpm until reaching $OD_{600}$ of approximately 1. Cells were harvested by centrifugation (4000× *g* for 10 min) and resuspended in PBS. A droplet of the resuspended cells was immobilized on a SuperFrost Ultra Plus slides (Invitrogen, Waltham, MA) for 45 min to allow the bacteria to adsorb. The unabsorbed cells were removed by washing with PBS, and the adsorbed cells were then blocked with 3% BSA for 45 min. Cells were then incubated with 100 µg ml⁻¹ Fn in PBS (Sigma-Aldrich, F0895) for 60 min at room temperature. The unbound Fn was removed by washing three times with PBS. The samples were then fixed with 4% paraformaldehyde for 30 min at room temperature. Immunolabeling was then performed as described above, anti-Fn mouse IgG (Sigma-Aldrich) diluted 1:100 in blocking buffer and the secondary antibody (anti-mouse IgG conjugated with Alexa Fluor 635, Goat IgG – Invitrogen) diluted 1:300 in blocking buffer. Cells were stained and prepared for imaging as described above.

## Preparation of polymer-coated surfaces

Quantification of interaction forces between Embp and Fn in its globular or fibrillated form would require that Fn was immobilized to a surface. We used a previously published model system (*Guerra et al., 2010*; *Rico et al., 2009*) to generate Fn-coated surfaces that displayed Fn in these two conformations, while the physicochemical properties of the underlying surface were very similar, namely PEA and PMA. Polymers of ethyl acrylate and methyl acrylate were synthesized from their monomers (99% pure, Sigma-Aldrich, Germany) using radical polymerization. Benzoin (98% pure, Sigma-Aldrich, Germany) was used as a photoinitiator with 1 wt % for PEA and 0.35 wt % for PMA. The polymerization reaction was allowed in Schlenk flasks exposing to ultraviolet light (portable UV lamp with light of 390–410 nm) up to the limited conversion of monomers (2 hr). Polymers were then dried to constant weight in a vacuum oven at 60°C for 12 hr. Both polymers were solubilized in toluene (99.8% pure, Sigma-Aldrich) to concentration of 6% w/v for PEA and 2.5% w/v for PMA. Two hours sonication in an ultrasonic bath at room temperature was used to make the polymer soluble. Glass slides (76 × 26 mm, Hounisen) were cleaned with sonication in ultrasonic bath for 15 min in acetone, ethanol, and Milli-Q water respectively, and then dried under nitrogen flow. A thin film of polymer solution was coated on clean slides using spin-coater (Laurell Technologies) with acceleration and velocity of 1000 rpm for 30 s. The spin-coated films were degassed in a desiccator for 30 min under vacuum and then put in a vacuum oven at 60°C for 2 hr to remove toluene.

## Fn adsorption to PMA and PEA

A hydrophobic marker (PAP pen – Sigma-Aldrich) was used to draw a small circle (around 0.5 cm square area) on the spin-coated slides. Fn from human plasma (Sigma-Aldrich, F0895) was dissolved in PBS at concentration of 20 µg ml⁻¹ and 100 µl sample was adsorbed on each slide for 1 hr at room temperature.

## AFM for imaging of Fn adsorbed to PMA and PEA

Experiments were conducted on three replicate samples with JPK Nanowizard IV (JPK, Germany) using HQ: CSC38/No Al (MikroMasch, San Jose, CA) and TR400PSA (Asylum Research, Santa Barbara, CA) cantilevers. We used the fluid mode of operation to visualize Fn adsorbed to PEA and PMA without

introducing artifacts from sample drying. The operating environment was controlled in a closed liquid chamber at 21°C with minimal evaporation. The operation parameters were set to optimize resolution with minimum possible damage or artifact from contaminations on the tip. Typically, scans were started with a large scan area of minimum $10 \times 10 \ \mu m^2$ with a rather low scan resolution of $64 \times 64$ pixels and a high scan rate >1 Hz. Once an area of interest was identified, a higher resolution image $256 \times 256$ or $512 \times 512$ pixels of a smaller scan are was acquired at lower scan speeds (<1 Hz). The acquired data was processed using Gwyddion open software (http://gwyddion.net/) for necessary corrections of tilt, etc.

## XPS of Fn adsorbed to PEA and PMA

A 100 µl of Fn (20 µg $ml^{-1}$) was adsorbed on a polymer spin-coated $1 \times 1 \ cm^2$ glass slides (these slides were cut manually in the chemistry lab workshop) for 1 hr, and samples were then washed three times with Milli-Q water and dried under $N_2$ flow. The chemical composition of the adsorbed layer was analyzed with a Kratos AXIS Ultra DLD instrument equipped with a monochromatic Al Kα X-ray source ($hv$=1486.6 eV). All spectra were collected in electrostatic mode at a take-off angle of 55° (angle between the sample surface plane and the axis of the analyzer lens). The spectra were collected at new spots on the sample ($n$=3, 1 replicate) and were charge corrected to the $C_{1s}$ aliphatic carbon binding energy at 285.0 eV, and a linear background was subtracted for all peak areas quantifications. Analyzer pass energy of 160 eV was used for compositional survey scans of $C_{1s}$, $O_{1s}$, $N_{1s}$, $Na_{1s}$, $Si_{2p}$, $Cl_{2p}$, $P_{2p}$, and $K_{2p}$. High-resolution scans of $C_{1s}$ and $N_{1s}$ elements were collected at an analyzer pass energy of 20 eV. Compositions and fits of the high-resolution scans were produced in CasaXPS. The data is presented in a table as an average and standard deviation of the three sample spots.

## FTIR analysis of Fn adsorbed to PEA and PMA

FTIR measurements were performed on a Bruker Vertex v70 with 128 scans per spectrum and a 7 mm diameter beam spot. The concentration of Fn (20 µg $ml^{-1}$) results in very small IR absorbances of the polymer layers, therefore, the spectra of stacks of eight coated CaF2 windows were measured simultaneously. For this, eight spin-coated CaF2 window surfaces with PMA and PEA were incubated for 1 hr with the 20 µg $ml^{-1}$ Fn solution in PBS prepared in $D_2O$ (d-PBS hereafter), after rinsing the surfaces with d-PBS and placed four sets of windows (spaced by 25 µm Teflon spacers that were filled by d-PBS, with the polymer and protein-coated sides submerged in the d-PBS) in a custom-made IR cell. The incubated sample spectra were background-corrected by subtracting the spectra of the same neat d-PBS loaded windows. Before subtraction, we (i) corrected for small differences in the overall transmission of the protein and background samples (due to e.g. small differences in the amount of scattering of the IR beam, which can become significant with eight consecutive windows) by subtracting the absorption at 7500 $cm^{-1}$, and (ii) corrected for small differences in the exact water layer thickness by scaling the spectra using a spectrally isolated absorption band of the $D_2O$ (the $v_1$ + $v_2$ combination band of the solvent's OD bending and stretching mode 1 at 3840 $cm^{-1}$) to determine the scaling factor. But there is no reason to assume that the water and polymer layers thicknesses are related. Therefore, the resulting background-corrected amide-I (1600–1700 $cm^{-1}$) PEA + Fn spectrum (*Figure 3*) still contains a tail of the 1733 $cm^{-1}$ ester peak, which is absent in the resulting PMA + Fn spectrum. This is (i) because there is approximately six times more PEA present than PMA (as indicated by a least-square fit that minimized the total intensity of the subtraction of the PMA from the PEA background spectra in the 1700–1760 $cm^{-1}$ region, see *Figure 3—figure supplements 4 and 5*), and (ii) because the Fn incubation results in a slight loss of polymer, which is impossible to compensate for well by subtraction of the polymer spectra, because the 1733 $cm^{-1}$ ester peak shape is affected by the presence of the protein (see *Figure 3—figure supplement 4d*). The broadening of this peak by Fn incubation is probably because the ester groups in contact with the protein are slightly shifted with respect to the more buried ester groups that are not changed by the protein adsorption, resulting in two subpeaks that are slightly offset in frequency. Even though the PMA layer appears to be thinner and/or less dense, it will probably still be composed of many monolayers (as indicated by the XPS measurements), so this difference in thickness is not expected to affect the protein's interfacial behavior.

## Quantification of bacterial attachment under flow

Ibidi sticky-slide VI 0.4 chambers (Ibidi, Germany) were glued to polymer-coated glass by using an equivalent mixture of silicon (DOWSIL 732 – Dow corning) and UV activating glue (Loctite 3106 Light Cure Adhesive). After flow cell assembly, 50 µl of Fn (20 µg ml$^{-1}$) dissolved in PBS was injected to the channel of a flow cell and allow to adsorb statically for 1 hr at room temperature. The unbound Fn was removed by a flow of PBS (6 ml hr$^{-1}$) using a syringe pump (Harvard Apparatus, Holliston, MA) for 15 min. Bacterial cells were subcultured from an overnight culture and grown for 6 hrs at 37°C and 180 rpm, harvested by centrifugation (4000× $g$ for 10 min), and resuspended in PBS to an OD$_{600}$ of 0.1. The cell suspension was flowed through the flow cell chamber at 3 ml hr$^{-1}$ for 2 hr at room temperature. The unbound cells were washed with PBS at 9 ml hr$^{-1}$ for 30 min. Attached bacteria were visualized by brightfield microscopy (Zeiss Axiovert A100, 20× objective) and counted. A minimum of five images were acquired per replicate, and a minimum of 200 cells were counted per replicate. Three replicate samples were analyzed, and individually grown bacterial cultures were used on each replicate.

The first experiment compared attachment of *S. carnosus* TM300 × pEmbp_5F and *S. carnosus* TM300 × pEmbp_9FG to PEA and PMA surfaces with and without Fn to investigate if the Fn-binding domains of Embp interacted selectively with the fibrillated form of Fn. The second experiment investigated which domain of Fn Embo interacted with. Previous studies had shown that that Embp binds to the F3 12th-14th domain of Fn (*Christner et al., 2010*), however, this experiment was only performed with recombinant fragments of Fn and not the full-length protein. We therefore investigated the role of this Fn domain in the attachment of bacteria via Embp. Flow cell experiments were carried out as described above, comparing attachment via F or FG to fibrillated Fn directly or after blocking for F3 12th-14th domain within IgG antibodies (anti-Fn, sc-18827, Santa Cruz Biotech). As a control, fibrillated Fn was blocked with IgG antibodies specific for another Fn domain (F3 5th domain). The unbound antibodies were removed with PBS (6 ml hr$^{-1}$, 15 min) before investigating bacterial attachment as described above.

The final experiment addressing attachment under flow compared the attachment of *S. epidermidis* 1585 Pxyl/tet *embp* and *S. epidermidis* Δ*embp*. The strains were inoculated from single colonies into BHI broth (amended with 5 µg ml$^{-1}$ erythromycin and 200 ng ml$^{-1}$ AHT for the Pxyl/tet *embp* strain) and grown overnight at 37°C, 180 rpm, harvested by centrifugation, and resuspended in PBS to OD$_{600}$=0.3, transferred to the syringe pump and passed through the flow cells at either 1 or 18 ml min$^{-1}$ flow rate for 1 hr followed by a 30 min PBS washing step of 6 or 36 ml min$^{-1}$, respectively. Attached cells were imaged by brightfield microscopy and by CLSM after staining with 20× SYBR Green II (Sigma-Aldrich).

## Hydrodynamic radius

*S. epidermidis* 1585 Pxyl/tet *embp* and *S. epidermidis* Δ*embp* were prepared as described above, transferred to cuvettes and analyzed by dynamic light scattering (Folded Capillary Zeta Cell, Malvern, PA). Measurements of surface charge and cell diameter were carried out using Zetasizer Nano (Malvern Panalytical) on three replicate samples prepared from individually grown overnight cultures.

## Single-cell force spectroscopy

SCFS measurements were conducted on Fn adsorbed in its globular conformation to PMA or its fibrillated conformation to PEA. For SCFS measurement, colloidal probes with 10 µm glass beads (SHOCON-BSG-B-5, Applied NanoStructures Inc, Mountain View, CA) were selected and coated by polymerizing a dopamine solution of 4 mg ml$^{-1}$ dopamine hydrochloride (99%, Sigma-Aldrich, H8502) in 10 mM Tris-HCl buffer at pH 8.5, and then calibrated in situ for single-cell attachment. *S. carnosus* TM300 expressing 5 F- and 9 FG-repeats were subcultured from overnight cultures and incubated for 6 hr in fresh media, harvested and resuspended in PBS as described above. A 100 µl drop of this solution was placed on a glass slide and incubated for 10 min, after which the unadsorbed bacteria were removed by rinsing with PBS. A colloidal probe was immersed and positioned on top of a single cell with the help of inverted optical microscope. The probe was made to contact a single cell for 5 min then retracted after the cell attachment. Once a cell was picked up (confirmed by optical microscopy), the substrate was changed to Fn-coated surfaces and SCFS was executed. The acquired

**Table 2.** Primers for recombinant Embp (rEmbp) cloning into pET302/NT-His.

| Primer | Sequence (5′ to 3′) |
|---|---|
| FG forward | AGAAGGAGATATACATATGCATCATCATCATCATCACGTGGAATTCGAAAACC TGTATTTTCAGGGCGGAGATCAAAAACTTCAAGATG |
| 1 FG reverse | TCCGATTATACCTAGGCTCGAATATCATCGATCTCGAGCGGAATTCTTAATGA AGATTTTGTTCAGC |
| 4 FG reverse | TCCGATTATACCTAGGCTCGAATATCATCGATCTCGAGCGGAATTCTTAATGT AAACTTTCTCTAGC |
| 15 FG reverse | TCCGATTATACCTAGGCTCGAATATCATCGATCTCGAGCGGAATTCTTAATT TAACGATGTTTCTGC |
| T7 Promotor | TAATACGACTCACTATAGGG |
| T7 Terminator | GCTAGTTATTGCTCAGCGG |

force-distance plots were processed using the Nanowizard's (JPK, Germany) own processing software. Experiments were conducted on two replicate samples (minimum 20 replicate force curves on each).

## Cloning and purification of F- and FG-repeats

Genomic DNA was extracted from *S. epidermidis* 1585 WT strain using the Qiagen DNA kit (Qiagen, Hilden, Germany) by following the instructions of the manufacturer. The only exception made in the kit protocol was that cells were lysed with 15 U of lysostaphin, which was added to buffer P1. The nucleotide sequence of 1, 4, and 15 repeats of the FG-repeats was amplified from genomic DNA using primers (*Table 2*) with Phusion High-Fidelity PCR Kit (NEB – E0553S). The PCR products were purified with GenElute PCR Clean-Up Kit (Sigma-Aldrich NA1020). The expression vector pET302/NT-His was digested with EcoR1 restriction enzyme (NEB R0101S) and run on a 1.5% agarose gel. The digested vector was purified from the gel using the GenElute Gel Extraction Kit (Sigma-Aldrich NA1111). The purified PCR product of each rEmbp was ligated with the digested vector in a ratio of 3:1 using the Gibson assembly ligation matrix mix (NEB E5510S). Each ligation reaction was incubated for 1 hr at 50°C. The ligated products were transformed into the chemically competent *Escherichia coli* strain (Top10). The colony PCR was performed with REDTaq ReadyMix (Sigma-Aldrich R2523) using the T7 promoter primer as forward and the T7 terminator primer as the reverse. Cells from each selected colony were grown overnight in LB with 100 µg ml$^{-1}$ ampicillin, and a plasmid miniprep was prepared using GeneJET Plasmid Miniprep Kit (Thermo Scientific K0702). Plasmids were sequenced with both T7 promoter and terminator primers by the Eurofins A/S (Hamburg, Germany). The plasmid of each Embp construct was transformed into an expression system (chemically competent *E. coli*, BL21-DE3). A single colony of the transformants was used to inoculate 2 l of LB with 100 µg ml$^{-1}$ ampicillin until the $OD_{600}$ of 0.6. For the overexpression, cells were induced with 1 M IPTG, and incubated for 16 hr on 28°C in shaking incubator at 180 rpm. Cells were harvested and lysed in binding buffer with sonication (30% amplitude, 15 s off, 15 s on) for 3 min on ice. After centrifugation, the supernatant was filtered with a 0.22 µm syringe filter and run on a nickel-nitrilotriacetic acid (Ni-NTA) column using ÄKTA Purifier-10 purification system. The column was washed with 5–8 column volumes, and the fusion proteins Embp was then eluted in fractions using elution buffer. Fractions of each rEmbp were pooled, concentrated with Amicon Ultra centrifugal filter tubes with a cut-off 3 kDa (Millipore Sigma UFC9003). Proteins were further purified with (HiTrap Q FF) column by anion exchange (IEX) chromatography using IEX binding and elution buffer, followed by size exclusion chromatography (SEC) with column (Superdex 200 Increase 10/300 GL) using MES buffer on ÄKTA Purifier-10 purification system.

**Table 3.** Buffer used for recombinant Embp (rEmbp) purification.

| Buffer name | Composition |
|---|---|
| Binding/lysis | 50 mM $K_2PO_4$, 500 mM NaCl, 400 mM imidazole, pH 7.4 |
| Ni-NTA elution | 50 mM $K_2PO_4$, 500 mM NaCl, 40 mM imidazole, pH 7.4 |
| IEX binding | 20 mM Bis-Tris propane, pH 6.0 |
| IEX elution | 20 mM Bis-Tris propane, 1 M NaCl, pH 6.0 |
| SEC | 50 mM MES, 150 mM NaCl, pH 6.0 |

After each column, the elution fractions were run on SDS-PAGE to check the purification quality. Buffers used for rEmbp purification are listed in *Table 3*.

## Single-molecule force spectroscopy

SMFS measurements, similar to SCFS, conducted on Fn adsorbed to either PEA and PMA. For the SMFS experiments, the probes were prepared by attaching rEmbp fragments of various lengths with the use of 6× His-NTA interaction. The procedure was similar to that of *Obataya et al., 2005*. In short, silicon probes were cleaned with ozone, and UV light then kept in one to one isopropyl alcohol and ethanol mixture overnight. Tips were rinsed in deionized water and air-dried, after which they were functionalized with 2% (3-mercaptopropyl) trimethoxysilane in EtOH for 30 min. Probes were then exposed to maleimide-C3-NTA in 50% DMF/100 mM Tris-HCl (pH7.5±0.1) overnight. Ten mM $NiCl_2$ was used to chelate the NTA groups on the tip, which was then dipped in bovine serum albumin (1 mg $ml^{-1}$ in PBS) to passivate the surface. The attachment of 6× His-tagged rEmbp fragments with 1, 4, or 15 repeating units of Fn-binding repeats was completed by 1 hr incubation of respective samples at room temperature. After a probe for each repeating unit was prepared, force-distance curves were collected on three replicate samples and processed, as mentioned above.

## Acknowledgements

This work was funded by the Carlsberg Foundation, Grant number CF16-0342. Cecilie Siem Bach-Nielsen is gratefully acknowledged for quantification of *S. epidermidis* attachment under high and low flow rates. TWG and SJR thank the Lundbeck Foundation for post doc fellowships.

## Additional information

### Funding

| Funder | Grant reference number | Author |
| --- | --- | --- |
| Carlsbergfondet | CF16-0342 | Nasar Khan<br>Hüsnü Aslan |
| Lundbeckfonden | Postdoctoral fellowship | Thaddeus Wayne Golbek<br>Steven Joop Roeters |

The funders had no role in study design, data collection and interpretation, or the decision to submit the work for publication.

### Author contributions

Nasar Khan, Steven Joop Roeters, Conceptualization, Formal analysis, Investigation, Methodology, Supervision, Validation; Hüsnü Aslan, Conceptualization, Formal analysis, Investigation, Methodology, Supervision, Validation, Visualization, Writing – original draft, Writing – review and editing; Henning Büttner, Investigation, Methodology, Resources, Supervision, Writing – original draft, Writing – review and editing; Holger Rohde, Visualization, Resources, Writing – original draft, Methodology, Validation; Thaddeus Wayne Golbek, Data curation, Conceptualization, Investigation, Methodology, Supervision, Validation; Sander Woutersen, Writing – original draft, Formal analysis, Investigation, Methodology, Validation; Tobias Weidner, Resources, Writing – original draft, Formal analysis, Methodology, Validation; Rikke Louise Meyer, Conceptualization, Funding acquisition, Investigation, Methodology, Project administration, Resources, Supervision, Validation, Writing – original draft, Writing – review and editing

### Author ORCIDs

Nasar Khan http://orcid.org/0000-0002-9196-9321
Hüsnü Aslan http://orcid.org/0000-0003-0471-5452
Henning Büttner http://orcid.org/0000-0002-5086-4961
Rikke Louise Meyer http://orcid.org/0000-0002-6485-5134

### Decision letter and Author response

Decision letter https://doi.org/10.7554/eLife.76164.sa1

Author response https://doi.org/10.7554/eLife.76164.sa2

## Additional files

### Supplementary files
• Transparent reporting form

### Data availability
Figure 1: Source data file contains original image files. Figure 2: Data are as shown in the figures. Figure 3: Source data file contains data and graph. Figure 4 and 5: Source data file contains force spectroscopy curves and processing information. Figure 5: Source data file contains the data and graph.

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
