## [Editor Report]

Bacteria must adhere to tissue to colonize and, in some cases, cause disease. Thus adherence is a key to understanding the pathogenesis of infectious diseases. This study uses a range of microscopic and biophysical measures to discover that Embp, a very long protein of repeating subunits, facilitates adherence of Staphylococcus epidermidis to damaged tissue or foreign bodies (e.g. catheters or implantable devices) even under high flow conditions such as those found in blood.

---

## [Decision Letter]

**Decision letter after peer review:**

Thank you for submitting your article "The giant staphylococcal protein Embp facilitates colonization of surfaces through Velcro-like attachment to fibrillated fibronectin" for consideration by *eLife*. Your article has been reviewed by 2 peer reviewers, and the evaluation has been overseen by a Reviewing Editor, Vaughn Cooper, and Wendy Garrett as the Senior Editor. The following individual involved in the review of your submission has agreed to reveal their identity: Paul Fey (Reviewer #1).

Essential revisions:

1) The flow cell experiments are very impactful. However, outstanding questions arise regarding if the investigators know what is facilitating adherence under low flow conditions? Does 1585 express Aap? There are a few experiments suggesting Aap may facilitate adherence to serum binding proteins.

2) Related to above: One of the important aspects of this paper is measuring bacterial adhesion to immobilized fibronectin under flow. It is important to determine how shear-stress impacts on the strength of the bond between Embp and fibronectin, which can be performed using AFM. For MSCRAMMs such as SdrG and ClfB, shear forces actually promote strong binding by simulating the final stage of the dock-latch-lock process, which should be addressed and may require experimentation.

3) It would aid the discussion if the authors could discuss the population biology of strains expressing Embp. I assume it is not on a mobilizable island…is it found primarily in ST2 strains, etc. I think this is important, have we selected strains that can adhere to foreign bodies? is Embp expression part of that selection process? The brief discussion related to the function of Embp and adherence to native, damaged tissue, which may have been the original function of this protein, is very effective.

*Reviewer #1 (Recommendations for the authors):*

There are a few concerns that I have, overall, I think the manuscript will be very impactful.

1. I would like to have seen a control in figure 1 documenting what the *S. aureus* phenotype is dependent upon. Is this phenotype Fibronectin binding protein-dependent?

2. Figure 3 and other aspects of the AFM. It seems there should be a S. carnosus TM300 control only without expression of Embp.

3. The flow cell experiments are very impactful. However, as a reader, I was really interested to know if the investigators knew what was facilitating adherence under low flow conditions? Does 1585 express Aap? There are a few experiments suggesting Aap may facilitate adherence to serum binding proteins.

4. It would aid the discussion if the group could discuss the population biology of strains expressing embp. I assume it is not on a mobilizable island…is it found primarily in ST2 strains, etc. I think this is important--have we selected for strains that can adhere to foreign bodies? is Embp expression part of that selection process? I was glad to see the brief discussion related to the function of Embp and adherence to native, damaged tissue, which may have been the original function of this protein.

Line 85-An advantage

Line 101, italicize S. epidermidis

*Reviewer #2 (Recommendations for the authors):*

–The role of AFM in elucidating the interactions between staphylococcal wall anchored proteins and ligands should be discussed in the introduction and discussion. This will place the current study in the context of the state of the art.

– The AFM experiments should be more detailed as exemplified by Dufrene's papers

The rationale for the application of AFM to study adhesin-ligand interactions should be explained simply for the non-expert. Three different approaches have been employed – non-parametric contact mode imaging of immobilized Fn (Figure 2), single cell force microscopy and single molecule force microscopy.

– The 3D structure of the Fn binding domains of Embp has been published by one of the co-authors of the current study. To advance the state of the art in a substantial way the molecular details of the binding interaction of Embp with the type III Fn domain should be provided or at least modelled.

– The graphic image should show a single Embp protein interacting with several Fn molecules bearing in mind that there is only one Embp binding domain in each Fn molecule. It would be interesting to know the stoichiometry of binding – to how many Fn molecules can a single Embp protein bind.

– A diagram of the structure of Fn and Embp should be included.

---

## [Author Response]

Essential revisions:1) The flow cell experiments are very impactful. However, outstanding questions arise regarding if the investigators know what is facilitating adherence under low flow conditions? Does 1585 express Aap? There are a few experiments suggesting Aap may facilitate adherence to serum binding proteins.

Previous work has demonstrated that *S. epidermidis* 1585 is Aap‐negative (Rohde et al., 2005,Molecular Microbiology 55(6):1883‐1895).

We expect that non‐specific interactions play a greater role in adhesion at low flow, and adhesion via other Fn‐ binding surface proteins may also play a relatively greater role under these conditions. The only other *S. epidermidis* adhesin that has been reported to bind to Fn is Aae, and our strain does contain the gene for Aae.

It is not possible to pinpoint which of the many possible adhesion mechanisms that are involved in adhesion of *S. epidermidis* to Fn‐coated surfaces at low flow. However, it is clear that Embp makes only a modest contribuation relative to other adhesion mechanisms under these conditions. We have elaborated the discussion, and the final paragraph in the Discussion section now reads:

“We show here that Embp makes a moderate contribution to attachment under low flow, leading to a 21% increase in the number of attached bacterial, while it is essential for attachment under high flow (Figure 7). Other than Embp, only the autolysin Aae has been reported to bind Fn based on in vitro studies of purified proteins [43], and S. epidermidis 1585 does contain this gene. It is possible that Aae and even non‐specific interaction forces contributed to adhesion of S. epidermidis at low flow, whereas only Embp attaches with sufficient force to prevent detachment at high flow.”

2) Related to above: One of the important aspects of this paper is measuring bacterial adhesion to immobilized fibronectin under flow. It is important to determine how shear-stress impacts on the strength of the bond between Embp and fibronectin, which can be performed using AFM. For MSCRAMMs such as SdrG and ClfB, shear forces actually promote strong binding by simulating the final stage of the dock-latch-lock process, which should be addressed and may require experimentation.

Thank you for this comment. We agree that the discussion should be elaborated with considerations of how the Embp‐Fn interaction compares with interaction mechanisms previously described for other bacterial adhesins and human ECM proteins.

The common binding mechanisms are the tandem‐zipper (e.g. between FnBP in *S. aureus* and Fn), and the “dock‐lock‐and‐latch” (DLL) mechanism between the Clf‐SdrG subfamily of MSCRAMMS and fibrinogen. The DLL mechanism is particularly interesting under flow, as the strength of the bond increases under mechanical stress (https://doi.org/10.1111/mmi.12663, https://doi.org/10.1016/j.jsb.2015.12.009, https://doi.org/10.1021/acs.langmuir.5b00360, https://doi.org/10.1021/acs.nanolett.9b03080).

The interaction between Embp and Fn, however, represents a novel mode of interaction between bacterial adhesins and host ECM proteins. Structural analyses (Büttner et al., 2020) do not suggest that it follows either the DLL or tandem zipper mechanisms, and it interacts with a different part of Fn, in volving FN12 near the C‐terminal of Fn which has never been implicated in interaction with bacterial Fn‐binding proteins before. Büttner et al., 2020 therefore argue that the Embp‐Fn interaction represents a novel principle in pathogen‐ECM interaction.

When investigating *S. epidermidis’* attachment to Fn under flow, we observed that the number of attached bacteria decreased at high flow, but the relative importance of Embp increased. We have re‐run the statistical analysis and discovered an error in our initial submission. Initially we deemed the contribution of Embp to attachment at low flow to not be significant, but a two‐tailed T‐test assuming unequal variance revealed a P value of 0.03. Hence, at low flow, Embp increased the number of attached bacteria by approximately 21%. Attachment at low flow was therefore primarily caused by other mechanisms, and one could argue that Embp contributes with approximately 20%. At high flow, there was no attachment of bacteria lacking Embp, making Embp solely responsible for attachment. The overall attachment rate at high flow was only 23% of the attachment rate at low flow. This number is consistent with the attachment rate one would predict if all other attachment mechanisms except Embp was eliminated. Our data therefore provides no indication that the attachment strength of an individual Embp protein is stronger at higher flow.

In preparation for adhesion experiments with *S.* carnosus expressing fractions of Embp, conducted initial adhesion experiments to Fn‐coated surfaces different flow rates (3, 6, and 12 ml/min), and we observed higher adhesion rates at the lowest flow rate, which was subsequently used for the experiments depicted in figure 4. The data was not recorded and only used to choose a flowrate for the subsequent experiments, but the observation strengthens our notion that Embp does not adhere more strongly at increasing flow rates.

We can speculate how flow might affect Embp‐mediated attachment. Considering that Embp contains 50 Fn‐binding repeats and that it is predicted to have a rod‐like structure (Büttner et al., 2020), the lateral force exerted by flow might stretch Embp across the Fn‐covered surface and increase the chance of multiple interactions. Such stretcing may occur at both flow‐rates used in our experiment, but it would be interesting to address how flow vs static conditions affect Embpmediated attachment in the future. Such an experiment requires a bacterial model that expressess full‐length Embp while lacking other Fn‐interacting mechanisms, and this model is not currently available. Embp is too large to be expressed at full length in a surrogate host, and development of such a model would therefore require knock‐out of all other adhesion mechanisms in the S. epidermidis strain carrying Embp. This work is therefore beyond the scope of our current paper.

We have elaborated the Discussion section of the manuscript with the points made above.

The last part of the Discussion section now reads:

“Previously published studies have shown that the binding force of some bacteria adhesins is promoted by external mechanical forces, such as shear stress when under flow, achieved in the “dock, lock and latch” mechanism observed for the Clf‐Sdr family of adhesins when binding to fibrinogen [44‐47]. However, our data do not suggest that increased shear stress would strengthen the interaction force between Embp and Fn, as the overall attachment rate of bacteria decreases by approximately 80% at the high flow rate. Furthermore, the structural analysis of Embp carried out by Büttner et al., [1] points out that the Embp represents a novel mode of interaction between bacterial adhesins and host ECM proteins. Considering the giant size of Embp, one can speculate that stretching of Embp due to lateral external mechanical forces will increase its contact area with surface‐organized Fn, and thereby increase the probability of successful binding events leading to adhesion. However, our data do not reveal whether this is the case. What we can conclude with certainty is that Embp is essential for attachment when host colonization is challenged by shear forces, and this observation suggests that Embp may be significant for biofilm formation in the cardiovascular system, e.g. on cardiovascular grafts. Future research in animal models will determine Embp’s role in colonization of endothelial cells and implants in the cardiovascular system.”

3) It would aid the discussion if the authors could discuss the population biology of strains expressing Embp. I assume it is not on a mobilizable island…is it found primarily in ST2 strains, etc. I think this is important, have we selected strains that can adhere to foreign bodies? is Embp expression part of that selection process? The brief discussion related to the function of Embp and adherence to native, damaged tissue, which may have been the original function of this protein, is very effective.

Recent work from the Rohde lab has demonstrated that while there is a specific ST signature related to invasive *S. epidermidis* strains, only few genetic elements were associated with invasive *S. epidermidis* isolates. Embp, however, is not enriched in invasive isolates, and can be found in invasive and colonizing isolates in almost equal percentages (Both et al., (2021) PloS Pathogens 17(2)). We have elaborated the Discussion section to include this information. The relevant paragraph now reads:

“The genome of *S. epidermidis* is highly variable, and not all isolates possess the same repertoire of genes for host colonization and biofilm formation [36]. Embp, however, is present in two thirds of *S. epidermidis* isolates from orthopedic device‐related infections [36] and 90 % of isolates from blood stream infections [9], which indicates some importance for its pathogenicity. In a recent study of adhesive vs invasive *S. epidermidis* isolates, Embp was not more frequent in any of the groups [28]. Hence, there is no indication of a general role of Embp for invasiveness vs colonization. We can speculate that strong attachment via a single protein like Embp could be particularly advantageous in locations where high shear forces make attachment difficult, such as in the blood stream. Biofilms generally do not form in blood vessels, unless there is an implant or a lesion on the endothelium. Infections like endocarditis often start with such a lesion [37], which makes the site susceptible to bacterial attachment. Fibrillated Fn forms on the surface of platelets in the early stages of wound healing [38], and perhaps the abundance of fibrillated Fn plays role in the elevated infection risk by *S. epidermidis* strains that carry Embp. The abundance of Embp among isolates from such infections remains to be investigated.”

Reviewer #1 (Recommendations for the authors):There are a few concerns that I have, overall, I think the manuscript will be very impactful.1. I would like to have seen a control in figure 1 documenting what the *S. aureus* phenotype is dependent upon. Is this phenotype Fibronectin binding protein-dependent?

The fibronectin‐binding proteins (FnBP) of *S. aureus* are capable of binding plasmafibronectin, i.e. the globular conformation of Fn. We therefore included it as a positive control to show that our labeled Fn could be detected on the surface of a bacterium, if it bound. As *S. aureus* has several Fn‐binding adhesins, we did not find it critical to dive into which of the adhesins were responsible for binding soluble Fn in this species. The purpose was to show that *S. epidermidis* does not, and for that we simply needed a positive control. *S. aureus* seemed like the obvious choice.

2. Figure 3 and other aspects of the AFM. It seems there should be a S. carnosus TM300 control only without expression of Embp.

*S. carnosus* has been used as a non‐adhesive genetic background to study adhesion properties of specific proteins in many studies, and it is this well characterized in this context (see e.g. Büttner et al., (2020). "A giant extracellular matrix binding protein of staphylococcus epidermidis binds surface‐immobilized fibronectin via a novel mechanism." mBio 11(5): 1‐9.)

In our study, we chose to block the hypothesized Embp‐binding domain in Fn to document the Embp‐specific interaction.

3. The flow cell experiments are very impactful. However, as a reader, I was really interested to know if the investigators knew what was facilitating adherence under low flow conditions? Does 1585 express Aap? There are a few experiments suggesting Aap may facilitate adherence to serum binding proteins.

Previous work has demonstrated that *S. epidermidis* 1585 is Aap‐negative (Rohde et al., 2005,Molecular Microbiology 55(6):1883‐1895).

We expect that non‐specific interactions play a greater role in adhesion at low flow, and adhesion via other Fn‐ binding surface proteins may also play a relatively greater role under these conditions. The only other *S. epidermidis* adhesin that has been reported to bind to Fn is Aae, and our strain does contain the gene for Aae.

We do not have data that pinpoint the different mechanisms involved in adhesion of *S. epidermidis* to Fn‐coated surfaces at low flow, but we have elaborated the discussion, and the final paragraph in the Discussion section now reads:

“We show here that Embp is required for attachment of *S. epidermidis* under high flow (Figure 7). Other than Embp, only the autolysin Aae has been reported to bind Fn based on in vitro studies of purified proteins [39], and *S. epidermidis* 1585 does contain this gene. It is possible that Aae and even non‐specific interaction forces contributed to adhesion of *S. epidermidis* at low flow, while the specific and strong interaction via Embp was decisive for successful attachment under more challenging conditions at high flow. The critical role of Embp under high flow points to a role for Embp in attachment of *S. epidermidis* e.g. to cardiovascular grafts. Future research in animal models will determine Embp’s role in host and implant colonization in the cardiovascular system.”

4. It would aid the discussion if the group could discuss the population biology of strains expressing embp. I assume it is not on a mobilizable island…is it found primarily in ST2 strains, etc. I think this is important--have we selected for strains that can adhere to foreign bodies? is Embp expression part of that selection process? I was glad to see the brief discussion related to the function of Embp and adherence to native, damaged tissue, which may have been the original function of this protein.

Recent work from the Rohde lab has demonstrated that while there is a specific ST signature related to invasive *S. epidermidis* strains, only few genetic elements were associated with invasive *S. epidermidis* isolates. Embp, however, is not enriched in invasive isolates, and can be found in invasive and colonizing isolates in almost equal percentages (Both et al., (2021) PloS Pathogens 17(2)). We have elaborated the Discussion section to include this information. The relevant paragraph now reads:

“The genome of *S. epidermidis* is highly variable, and not all isolates possess the same repertoire of genes for host colonization and biofilm formation [36]. Embp, however, is present in two thirds of *S. epidermidis* isolates from orthopedic device‐related infections [36] and 90 % of isolates from blood stream infections [9], which indicates some importance for its pathogenicity. In a recent study of adhesive vs invasive *S. epidermidis* isolates, Embp was not more frequent in any of the groups [28]. Hence, there is no indication of a general role of Embp for invasiveness vs colonization. We can speculate that strong attachment via a single protein like Embp could be particularly advantageous in locations where high shear forces make attachment difficult, such as in the blood stream. Biofilms generally do not form in blood vessels, unless there is an implant or a lesion on the endothelium. Infections like endocarditis often start with such a lesion [37], which makes the site susceptible to bacterial attachment. Fibrillated Fn forms on the surface of platelets in the early stages of wound healing [38], and perhaps the abundance of fibrillated Fn plays role in the elevated infection risk by *S. epidermidis* strains that carry Embp. The abundance of Embp among isolates from such infections remains to be investigated.”

Line 85-An advantage

OK

Line 101, italicize S. epidermidis

OK

Reviewer #2 (Recommendations for the authors):–The role of AFM in elucidating the interactions between staphylococcal wall anchored proteins and ligands should be discussed in the introduction and discussion. This will place the current study in the context of the state of the art.

We thank the reviewer for pointing out at the non‐expert audiences’ perspective and understanding. As the reviewer suggested, we have added the rationale behind using advanced atomic force microscopy methods to investigate the surface morphology and quantitative biomolecular interactions and referenced high‐impact publications for example by Dufrene *et al.,* as to support the rationale. The quoted texts below are added to the manuscript’s introduction and Discussion sections. The last paragraph of the Introduction now reads:

“In this study, we investigate Embp’s interaction with Fn. Fn circulates in bodily fluids in a compact globular form [13], while fibrillated Fn contributes to the assembly of the extracellular matrix of tissue [14]. It is the stretching of Fn upon interacting with cell surface integrins, which exposes selfbinding domains and trigger Fn fibrillation. This mechanism ensures that Fn only fibrillates in the extracellular matrix of tissue and not in the blood stream [15]. We hypothesize that *S. epidermidis* interacts selectively with fibrillated Fn, and that a fibrillated ligand provides an opportunity for a multivalent interaction with the many repetitive F and FG repeats of the Embp. Using a model system of polymer‐coated surfaces that facilitate Fn adsorption in either globular or fibrillated conformation, we probed Embp’s interaction with Fn by flow cell and advanced Atomic Force Microscopy (AFM) experiments. The AFM can be used as a tactile tool for biological samples, which employs a cantilever with a sharp tip scanned over a sample with low forces in air or liquid environments[16, 17]. We used a mode of AFM in which the sharp tip intermittently contacts the surface while scanning over it to create a 3D real‐space image. Doing so enabled us to reveal the globular and fibrillar Fn conformations on polymer surfaces. Beyond the surface morphology, AFM can be used to measure the quantitative forces acting between the tip and the sample surface [18]. The latter, provides insights in many biological binding events by the modification of the probe, for example by attaching a single‐cell at the end of the cantilever instead of a sharp tip [19]. Attaching a cell or proteins on the probe and using it to measure quantitative forces on the sample of interest are referred as Single Cell or Single Molecule Force Spectroscopy which enables the observation of the nature of biomolecular binding events and their dynamics. Using native and recombinant Embp in a series of analyses at the population‐, single‐cell‐, and single‐molecule levels, we confirmed that Embp selectively interacts with fibrillated Fn. The interaction is a Velcro‐like mechanism where multiple binding‐domains must interact simultaneously to facilitate strong attachment. Such strong attachment via a single protein is particularly beneficial under high sheer stress, such as in the vascular system, and it was exactly under these conditions that Embp gave the cells an advantage. Embp homologs are present in other important pathogens capable of biofilm formation in the vascular system, and our study reveals a mechanism for how bacteria accomplish this feat.”

– The AFM experiments should be more detailed as exemplified by Dufrene's papersThe rationale for the application of AFM to study adhesin-ligand interactions should be explained simply for the non-expert. Three different approaches have been employed – non-parametric contact mode imaging of immobilized Fn (Figure 2), single cell force microscopy and single molecule force microscopy.

This valuable comment by the reviewer aligns well with the previous one, therefore we have addressed them simultaneously. Please see the reply to the above comment.

– The 3D structure of the Fn binding domains of Embp has been published by one of the co-authors of the current study. To advance the state of the art in a substantial way the molecular details of the binding interaction of Embp with the type III Fn domain should be provided or at least modelled.

We validate the interaction of F and FG repeats with the FNIII(12‐14) domain using full‐length Fn in this study. While it would be interesting to study the molecular details of the interaction further, it is beyond the scope of our current study.

– The graphic image should show a single Embp protein interacting with several Fn molecules bearing in mind that there is only one Embp binding domain in each Fn molecule. It would be interesting to know the stoichiometry of binding – to how many Fn molecules can a single Embp protein bind.

We agree with the reviewer’s comment and have added a new figure to the manuscript (Figure 1). The stochiometry cannot be determined from our data, and the figure is therefore not quantitative but illustrates multiple interactions between Fn and Embp.

– A diagram of the structure of Fn and Embp should be included.

We have amended such a figure to the manuscript (Figure 1)